# Phosphorylation acts positively and negatively to regulate MRTF-A subcellular localisation and activity

Richard Panayiotou[1], Francesc Miralles[1†‡], Rafal Pawlowski[1†§], Jessica Diring[1], Helen R Flynn[2], Mark Skehel[2¶], Richard Treisman[1]*

[1]Signaling and Transcription Group, Francis Crick Institute, London, United Kingdom; [2]Mass Spectrometry Science Technology Platform, Francis Crick Institute, London, United Kingdom

*For correspondence: Richard.
Treisman@Crick.ac.uk

[†]These authors contributed
equally to this work

Present address:
[‡]Cardiovascular and Cell
Sciences Research Institute and
Institute for Medical and
Biomedical Education, St
George's, University of London,
London, United Kingdom;
[§]inVentiv Health, Kuesnacht,
Switzerland; [¶]MRC Laboratory of
Molecular Biology, Cambridge,
United Kingdom

Competing interests: The
authors declare that no
competing interests exist.

Reviewing editor: Roger J
Davis, University of
Massachusetts Medical School,
United States

**Abstract** The myocardin-related transcription factors (MRTF-A and MRTF-B) regulate cytoskeletal genes through their partner transcription factor SRF. The MRTFs bind G-actin, and signal-regulated changes in cellular G-actin concentration control their nuclear accumulation. The MRTFs also undergo Rho- and ERK-dependent phosphorylation, but the function of MRTF phosphorylation, and the elements and signals involved in MRTF-A nuclear export are largely unexplored. We show that Rho-dependent MRTF-A phosphorylation reflects relief from an inhibitory function of nuclear actin. We map multiple sites of serum-induced phosphorylation, most of which are S/T-P motifs and show that S/T-P phosphorylation is required for transcriptional activation. ERK-mediated S98 phosphorylation inhibits assembly of G-actin complexes on the MRTF-A regulatory RPEL domain, promoting nuclear import. In contrast, S33 phosphorylation potentiates the activity of an autonomous Crm1-dependent N-terminal NES, which cooperates with five other NES elements to exclude MRTF-A from the nucleus. Phosphorylation thus plays positive and negative roles in the regulation of MRTF-A.

## Introduction

The myocardin-related transcription factors (MRTFs, also called MKLs), act in partnership with serum response factor (SRF) to control cytoskeletal gene expression in development, morphogenesis, and cell migration (*Arsenian et al., 1998*; *Esnault et al., 2014*; *Schratt et al., 2002*). The MRTFs are G-actin binding RPEL proteins (*Miralles et al., 2003*). Their activity responds to Rho GTPase-induced changes in cellular G-actin concentration, thereby coupling cytoskeletal gene expression to cytoskeletal dynamics (*Esnault et al., 2014*; *Olson and Nordheim, 2010*; *Posern and Treisman, 2006*). The N-terminal MRTF regulatory RPEL domain assembles tri- and pentameric complexes with G-actin (*Hirano and Matsuura, 2011*; *Mouilleron et al., 2011*). MRTF mutants that cannot bind G-actin are constitutively active (*Miralles et al., 2003*; *Vartiainen et al., 2007*), as is myocardin, the muscle-specific founder of the MRTF family (*Wang et al., 2001*), which cannot bind G-actin owing to sequence changes in its RPEL domain (*Guettler et al., 2008*).

Although predominantly cytoplasmic in unstimulated cells, the MRTFs continuously shuttle through the nucleus. Signal-induced G-actin depletion decreases MRTF-actin interaction in both the nucleus and cytoplasm, thereby promoting MRTF nuclear accumulation (*Guettler et al., 2008*; *Miralles et al., 2003*; *Vartiainen et al., 2007*). G-actin competes with importin αβ for binding to an NLS within the RPEL domain (*Hirano and Matsuura, 2011*; *Mouilleron et al., 2011*; *Pawłowski et al., 2010*). It cooperates with the Crm1 exportin in MRTF nuclear export (*Vartiainen et al., 2007*). Two Crm1-dependent MRTF NES elements have been reported, but the

mechanism of this cooperation remains unresolved (*Hayashi and Morita, 2013*; *Muehlich et al., 2008*). Confinement of the MRTFs to the nucleus in resting cells fails to activate target genes unless the G-actin/MRTF interaction is disrupted, implicating nuclear G-actin directly in transcriptional control (*Baarlink et al., 2013*; *Vartiainen et al., 2007*).

Growth factor stimulation also acts via Rho- and ERK-dependent pathways to induce MRTF phosphorylation (*Kalita et al., 2006*; *Miralles et al., 2003*; *Muehlich et al., 2008*), and proteomic studies in various cell types have identified multiple phosphorylation sites (*Badorff et al., 2005*; *Gnad et al., 2011*, *2007*; *Muehlich et al., 2008*). However, the extent and function of phosphorylation of the MRTFs, and of myocardin, remain largely unclear. For example, MRTF-A phosphorylation by ERK at residue S549 (numbering for 'fl' form, *Miralles et al. 2003)* promotes nuclear export by facilitating G-actin binding (*Muehlich et al., 2008*), but GSK3 phosphorylation of myocardin, and possibly MRTF-A, at the corresponding site and its neighbours promotes its degradation (*Badorff et al., 2005*; *Charbonney et al., 2011*; *Xie et al., 2009*). ERK phosphorylation of C-terminal sites in myocardin also appears to reduce its ability to activate transcription (*Taurin et al., 2009*).

Here, we show that Rho-dependent MRTF phosphorylation reflects its nuclear accumulation and dissociation from G-actin, and identify multiple sites for MRTF phosphorylation, which contribute to transcriptional activation. Phosphorylation of S98, within the RPEL domain, inhibits G-actin/MRTF complex formation and promotes nuclear import, while phosphorylation of S33 potentiates activity of a previously unidentified Crm1-dependent NES N-terminal to the RPEL domain. We identify four other NES elements in MRTF-A, which functionally cooperate with each other, and with the N-terminal phosphorylations, to control MRTF-A subcellular localisation in resting cells.

## Results

### G-actin binding negatively regulates MRTF phosphorylation

Using mouse NIH3T3 fibroblasts, we first investigated the relationship between transcriptional activation by MRTF-A and its phosphorylation, as assessed by its reduced mobility in SDS-PAGE (*Miralles et al., 2003*). Although this approach detects only those phosphorylations that significantly alter mobility and which occur at high stoichiometry, and cannot be used to unambigously identify phosphorylation sites, it does provide a useful way to investigate the global factors controlling MRTF-A phosphorylation. Significant MRTF-A phosphorylation persisted for 6–8 hr after serum stimulation, mirroring the nuclear retention time of MRTF-A in this system (*Vartiainen et al., 2007*), while transient transcriptional induction of MRTF-A target genes had declined to basal levels within 2 hr (*Figure 1A*). Serum washout, or Latrunculin B (LatB) treatment, which restore cytoplasmic localisation of MRTF-A (*Vartiainen et al., 2007*), resulted in the rapid return of MRTF-A phosphorylation to prestimulation levels (*Figure 1B*). We previously showed that serum-induced MRTF phosphorylation comprises both ERK- and Rho-dependent components (*Figure 1—figure supplement 1A*, *Miralles et al., 2003*), and direct activation of ERK signalling by TPA or Raf activation also induced MRTF-A phosphorylation (*Figure 1C*; *Figure 1—figure supplement 1B*). In contrast, although the actin-binding drug cytochalasin D (CD), which directly activates MRTF by preventing its association with G-actin, also induced MRTF-A phosphorylation, this occurred independently of ERK (*Figure 1C*).

The ability of CD to stimulate MRTF-A phosphorylation led us to test whether it could be induced by alterations in actin dynamics alone. Serum-induced MRTF nuclear accumulation in NIH3T3 cells is dependent on Rho signalling to the formin mDia1, which nucleates F-actin assembly (*Copeland and Treisman, 2002*; *Miralles et al., 2003*). Expression of an activated mDia1 derivative, which promotes MRTF-A nuclear accumulation, also induced its phosphorylation (*Figure 1D*, *Miralles et al., 2003*). In contrast, relocalisation of MRTF-A to the nucleus of resting cells by its fusion to an heterologous NLS, or treatment of the cells with Leptomycin B, did not induce phosphorylation unless the cells were simultaneously stimulated with CD or FCS (*Figure 1E,F*). This suggests that phosphorylation requires dissociation of G-actin from MRTF-A, and that in the context of these experiments, phosphorylation is inhibited by interaction between nuclear G-actin and MRTF-A. Indeed, the MRTF-A derivative MRTF-A-123-1A, which cannot bind actin and is constitutively nuclear (*Vartiainen et al., 2007*), exhibited elevated levels of phosphorylation in resting cells and was not sensitive to mDia expression (*Figure 1D*). Myocardin also does not bind actin and is constitutively nuclear; however,

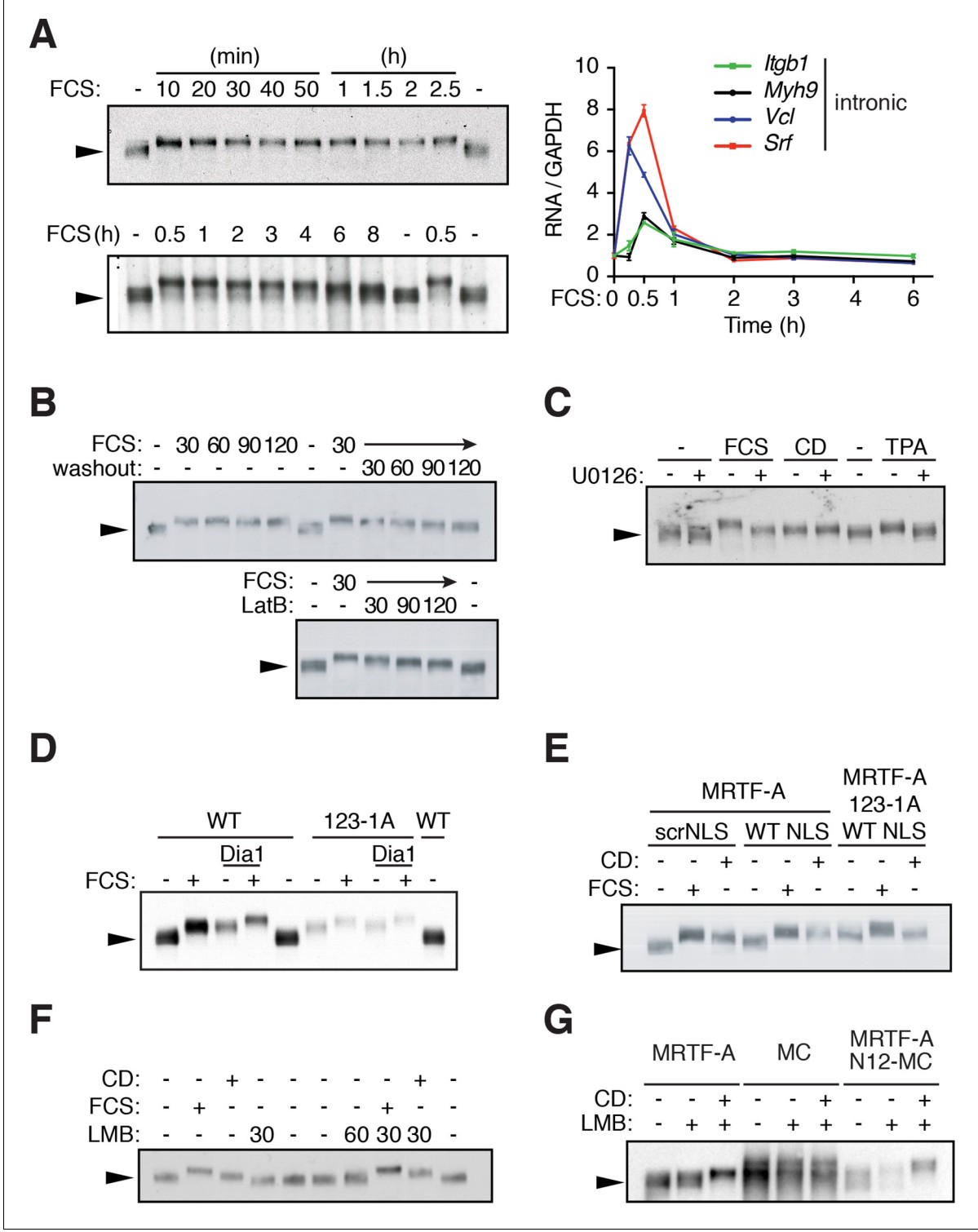

**Figure 1.** G-actin controls MRTF phosphorylation. (**A**) MRTF-A phosphorylation persists after transcriptional shutoff. NIH3T3 cells were stimulated with 15% FCS for the indicated times. Left, Phosphorylation of MRTF-A (arrowed) was detected as reduced mobility in SDS-PAGE by immunoblot. Right, transcription of MRTF-A target genes, analysed by qPCR using intronic primers, normalised to GAPDH, and expressed relative to basal conditions. Data are from three independent experiments, plotted as mean ± SEM. (**B**) Cells were stimulated with 15% FCS as indicated; 30 min later the medium was replaced with 0.5% FCS or 1 μM Latrunculin B (LatB) was added, and endogenous MRTF-A phosphorylation monitored by SDS-PAGE. (**C**) Cells were stimulated with 15% FCS, 2 μM CD or 100 ng/ml TPA with and without 10 μM U0126. (**D**) MRTF-A phosphorylation is induced by mDia1 coexpression or disruption of G-actin binding. Cells expressing exogenous mouse MRTF-A or MRTF-A-123-1A, which cannot bind G-actin (**Mouilleron et al., 2008**;

*Figure 1 continued on next page*

*Figure 1 continued*

*Vartiainen et al., 2007*) and constitutively active mDia1 were treated with 15% FCS and MRTF-A phosphorylation monitored by SDS-PAGE. (E) Nuclear localisation of MRTF-A is not sufficient for its phosphorylation. Cells expressing MRTF-A linked to an intact or scrambled SV40 NLS were stimulated with 15% FCS or 2 µM CD for 30 min. (F) Cells were stimulated with 15% FCS, 2 µM CD or 50 nM Leptomycin B (LMB); MRTF-A phosphorylation was monitored by SDS-PAGE and immunoblotting. (G) Cells expressing MRTF-A, Myocardin (MC) or MRTF-A N12-MC, in which MC N-terminal sequences including RPEL1 and RPEL2 are replaced by those of MRTF-A (*Guettler et al., 2008*), were treated with 50 nM LMB with or without 2 µM CD as indicated. For effect on reporter gene activity, see *Figure 1—figure supplement 1C*.

The following figure supplement is available for figure 1:

**Figure supplement 1.** Regulation of MRTF-A phosphorylation.

replacement of its N-terminal sequences with those of MRTF-A was sufficient to restore G-actin binding and regulation (*Figure 1—figure supplement 1C*, *Guettler et al., 2008*). Consistent with this, the MRTF-A/MC fusion protein exhibited CD-induced phosphorylation similar to that seen in MRTF-A (*Figure 1G*). Finally, phosphorylation of the NLS mutant MRTF-A B2A, which cannot accumulate in the nucleus following serum stimulation (*Pawłowski et al., 2010*) could be induced by serum, but not CD, and this was entirely ERK-dependent (*Figure 1*, *Figure 1—figure supplement 1D*). Thus, dissociation of G-actin from MRTF-A underlies its ERK-independent phosphorylation (see Discussion).

## Phosphorylation potentiates transcriptional activation by MRTF-A

Previous studies have identified a number of phosphorylation sites on myocardin family proteins, including MRTF-A (*Badorff et al., 2005*; *Gnad et al., 2011*, *2007*; *Muehlich et al., 2008*). We generated tetracycline-inducible cell lines expressing FLAG-tagged MRTF-A and analysed its phosphorylation following serum stimulation. At least 26 high confidence phosphorylation sites were identified by mass spectrometric analysis of phosphopeptides derived from cleavage with trypsin, chymotrypsin and AspN and by generation of phosphorylation site-specific antisera (*Table 1*, *Figure 2A*). The majority of these were at S/T-P motifs, and many of these were identified as ERK targets in preliminary analyses in which recombinant fragments of MRTF-A were phosphorylated by ERK in vitro. These results confirm and extend the catalogue of MRTF-A phosphorylation sites detected in mouse fibroblasts using high-throughput proteomics (*Figure 2—figure supplement 1*, *Hsu et al., 2011*; *Wu et al., 2012*; *Yu et al., 2011*).

Phospho-specific antibodies were generated against 13 phospho-sites. Apart from S98, all exhibited a basal level of phosphorylation in unstimulated cells; upon serum stimulation, phosphorylation of S98, S231, S663 and S744 increased substantially within 15 min, while that of S33, S248, S605, S708, S775, S785, S867, T879 and S897/T899 increased to a lesser extent (*Figure 2—figure supplement 2A,B,C,D*). In contrast, CD or TPA stimulation induced a gradual accumulation at all analysed sites (*Figure 2—figure supplement 2A,C*).

Given the sensitivity of MRTF-A phosphorylation to inhibition of the Ras-ERK pathway, we generated an MRTF-A derivative, MRTF26A, in which all phosphorylated S/T-P motifs were substituted by alanine (*Figure 2A*). MRTF26A exhibited increased and more uniform mobility than the wild-type protein MRTF-A in SDS-PAGE (*Figure 2B*). In contrast to wild-type MRTF-A, serum stimulation or expression of activated mDia1 did not alter the mobility of MRTF26A in SDS-PAGE, consistent with decreased phosphorylation (*Figure 2B*). Moreover, unlike wild-type MRTF-A, disruption of the G-actin-binding sites in the RPEL domain of MRTF26A did not detectably induce phosphorylation as assessed by the SDS-PAGE assay (*Figure 2B*). The MRTF26A mutant thus exhibits greatly reduced susceptibility to phosphorylation.

Like wild-type MRTF-A, MRTF26A was cytoplasmic in resting cells and accumulated in the nucleus following serum stimulation (*Figure 2C*). Nevertheless, although overexpression of either wild-type MRTF-A or MRTF26A potentiated activity of the SRF reporter gene similarly in unstimulated cells, MRTF26A exhibited significantly reduced activity following serum or CD stimulation (*Figure 2D*). Since this reduction was also observed upon expression of MRTF26A-123-1A, which is constitutively nuclear, it cannot reflect defective nuclear accumulation (*Figure 2E*). Taken together, these results suggest that phosphorylation potentiates transcriptional activation by MRTF-A.

**Table 1.** MRTF-A phosphorylation sites. NIH3T3 cells stably expressing Flag-MRTF-A were stimulated with serum for 30 min, immunoprecipitated with Flag antibody and fractionated by SDS-PAGE in triplicate. The MRTF-A bands were excised, digested with Trypsin, AspN or Chymotrypsin and the digests analysed with and without titanium dioxide enrichment on the LTQ Orbitrap Discovery. Fixed modifications were set as Carbamidomethyl (C) and variable modifications as Oxidation (M) and Phospho (STY). The estimated false discovery rate was set to 1% at the peptide, protein and site level. For all enzymatic digests, a maximum of two missed cleavages was allowed. Raw data were processed in MaxQuant (version 1.3.0.5) for peptide and protein identification, searching against the canonical sequence of *Mus Musculus* from the Uniprot KB, release 2012_08. The summary table was generated from the MaxQuant output file PhosphoSTY Sites.txt, an FDR-controlled site-based table compiled by MaxQuant from the relevant information about the identified peptides. Modified residues are shown in the left columns, with residue positions for MRTF-A(fl) and MRTF-A(met) indicated. In the sequence window the central residue is the putative modified site, shown in bold. For each site, the MS data are summarised as the best peptide Posterior Error Probability (PEP) score and the localisation probability, and/or the availability of a phospho-site specific antibody indicated. Blue shading indicates sites denoted reliable on the basis of their satisfying PEP <1.0E-06 and Localisation Probability >0.75 (>0.5 for repeated phosphoacceptors), confirmation by phosphosite-specific antibody, or manual peptide identification for residues 587 and 601 (peptides TQLTLQAS(ph)PL and AAS(ph)C(carb)C(carb)LS(ph)PGAR respectively, shown in blue in the sequence window). Residues mutated to Alanine in MRTF26A are shown in red. Peptides used for immunisation are underlined; those used for S98, S231, S663, S867 and ST897/9 were synthesised with C-terminal cysteines.

| Amino acid | MRTF-A(FL) | MRTF-A(Met) | Sequence | PEP score | Localizn prob. | Phospho-antibody |
|---|---|---|---|---|---|---|
| S | 33 | n/a | ENDDEPVLLSL<u>SAAP**S**PQSE</u>AVANELQELSL | 1.39E-10 | 0.993037 | pAb33 |
| S | 98 | 6 | TREELVSQGIM<u>PPLK**S**PAAFHEQ</u>RRSLERAR | 1.16E-06 | 1 | pAb98 |
| S | 206 | 114 | IIVGQVNYPKVADSS**S**FDEDSSDALSPEQPA | 0.0211238 | 0.310963 | |
| S | 211 | 119 | VNYPKVADSSSFDED**S**SDALSPEQPASHESQ | 0.000127434 | 0.799217 | |
| S | 212 | 120 | NYPKVADSSSFDEDS**S**DALSPEQPASHESQG | 0.0059612 | 0.371831 | |
| S | 216 | 124 | VADSSSFDEDSSDAL**S**PEQPASHESQGSVPS | 1.67E-30 | 1 | |
| S | 231 | 139 | SPEQPASHES<u>QGSVP**S**PLESRV</u>SDPLPSATS | 1.69E-23 | 1 | pAb231 |
| S | 248 | 156 | LESRVSDPL<u>PSATSI**S**PTQVLSQLP</u>MAPDPG | 4.25E-16 | 0.999792 | pAb248 |
| T | 402 | 310 | APPKPSAETPGSSAP**T**PSRSLSTSSSPSSGT | 0.000114318 | 0.993969 | |
| S | 406 | 314 | PSAETPGSSAPTPSR**S**LSTSSSPSSGTPGPS | 9.19E-10 | 0.559961 | |
| S | 408 | 316 | AETPGSSAPTPSRSL**S**TSSSPSSGTPGPSGL | 2.23E-11 | 0.676126 | |
| T | 409 | 317 | ETPGSSAPTPSRSLS**T**SSSPSSGTPGPSGLA | 2.49E-09 | 0.497067 | |
| S | 412 | 320 | GSSAPTPSRSLSTSS**S**PSSGTPGPSGLARQS | 2.23E-11 | 0.99797 | |
| S | 415 | 323 | APTPSRSLSTSSSPS**S**GTPGPSGLARQSSTA | 4.58E-06 | 0.524459 | |
| T | 417 | 325 | TPSRSLSTSSSPSSG**T**PGPSGLARQSSTALA | 2.34E-09 | 0.998496 | |
| S | 428 | 336 | PSSGTPGPSGLARQS**S**TALAAKPGALPANLD | 0.000953078 | 0.745616 | |
| T | 429 | 337 | SSGTPGPSGLARQSS**T**ALAAKPGALPANLDD | 0.000953078 | 0.493371 | |
| S | 480 | 388 | KTELIERLRAYQDQV**S**PAPGAPKAPATTSVL | 2.99E-06 | 1 | |
| S | 541 | 449 | MVVATVTSNGMVKFG**S**TGSTPPVSPTPSERS | 1.93E-11 | 0.96512 | |
| T | 542 | 450 | VVATVTSNGMVKFGS**T**GSTPPVSPTPSERSL | 3.23E-07 | 0.899805 | |
| S | 544 | 452 | ATVTSNGMVKFGSTG**S**TPPVSPTPSERSLLS | 3.78E-08 | 0.916768 | |
| T | 545 | 453 | TVTSNGMVKFGSTGS**T**PPVSPTPSERSLLST | 2.23E-12 | 0.994425 | |
| S | 549 | 457 | NGMVKFGSTGSTPPV**S**PTPSERSLLSTGDEN | 2.23E-12 | 0.999632 | |
| T | 551 | 459 | MVKFGSTGSTPPVSP**T**PSERSLLSTGDENST | 2.23E-12 | 0.94872 | |
| S | 553 | 461 | KFGSTGSTPPVSPTP**S**ERSLLSTGDENSTPG | 3.78E-08 | 0.998455 | |
| S | 559 | 467 | STPPVSPTPSERSLL**S**TGDENSTPGDAFGEM | 0.00405521 | 0.999782 | |
| T | 560 | 468 | TPPVSPTPSERSLLS**T**GDENSTPGDAFGEMV | 0.00405521 | 0.499997 | |
| S | 565 | 473 | PTPSERSLLSTGDEN**S**TPGDAFGEMVTSPLT | 0.000480998 | 0.837039 | |
| S | 577 | 485 | DENSTPGDAFGEMVT**S**PLTQLTLQASPLQIV | 5.33E-09 | 0.999998 | |
| S | 587 | 495 | GEMVTSPLT<u>QLTLQA**S**PLQIVKEEGARAASC</u> | 0.00523736 | 1 | |
| S | 601 | 509 | ASPLQIVKEEGAR<u>AA**S**CCLSPGAR</u>AELEGLD | 0.000780733 | 1 | |

*Table 1 continued on next page*

Table 1 continued

| Amino acid | MRTF-A(FL) | MRTF-A(Met) | Sequence | PEP score | Localizn. prob. | Phospho-antibody |
|---|---|---|---|---|---|---|
| S | 605 | 513 | QIVKEEGARAASCCL**S**PGARAELEGLDKDQM | 2.27E-13 | 1 | pAb605 |
| S | 662 | 570 | LQLEQQKRAQQPAPA**S**SPVKRESGFSSCQLS | 1.02E-08 | 0.998348 | |
| S | 663 | 571 | QLEQQKRAQQPAPASS**S**PVKRESGFSSCQLSC | 0.000148519 | 0.998905 | pAb663 |
| S | 708 | 616 | VPTTNHGDTQAPAPE**S**PPVVVKQEAGPPEPD | 1.17E-24 | 1 | pAb708 |
| S | 744 | 652 | QLLLGSQGTSFLKRV**S**PPTLVTDSTGTHLIL | n/a | n/a | pAb744 |
| S | 775 | 683 | TVTNKSADGPGLPAGS**S**PQQPLSQPGSPAPGP | 7.59E-16 | 1 | pAb775 |
| S | 781 | 689 | ADGPGLPAGSPQQPL**S**QPGSPAPGPPAQMDL | 9.94E-13 | 0.998507 | |
| S | 785 | 693 | GLPAGSPQQPLSQPGS**S**PAPGPPAQMDLEHPP | 3.03E-13 | 0.999999 | pAb785 |
| S | 867 | 775 | SADFKEPPSLPGKEK**S**PPAAAAYGPPLTPQP | 0.000807143 | 1 | pAb867 |
| T | 879 | 787 | KEKSPPAAAAYGPPL**T**PQPSPLSELPQAAPP | 0.0101532 | 1 | pAb879 |
| S | 883 | 791 | PPAAAAYGPPLTPQP**S**PLSELPQAAPPPGSP | 1.36E-08 | 1 | |
| S | 897 | 807 | PSPLSELPQAAPPPG**S**PTLPGRLEDFLESST | 1.92E-15 | 0.999573 | pAb897/9 |
| T | 899 | 809 | PLSELPQAAPPPGSP**T**LPGRLEDFLESSTGL | n/a | n/a | pAb897/9 |
| S | 941 | 849 | PLSLIDDLHSQMLSS**S**AILDHPPSPMDTSEL | 0.000584126 | 0.569645 | |
| S | 949 | 857 | HSQMLSSSAILDHPP**S**PMDTSELHFAPEPSS | 8.45E-22 | 1 | |

## Phorbol esters act positively and negatively to regulate MRTF-A

ERK signalling has previously been implicated in MRTF activation in neuronal cells (*Kalita et al., 2006*). In NIH3T3 cells, the phorbol ester TPA, which activates ERK through cooperation between Protein Kinase C (PKC) family members and RasGRP (see *Griner and Kazanietz, 2007* for review), induced transcription of MRTF-SRF target genes in an ERK-dependent manner (*Figure 3A*, *Gineitis and Treisman, 2001*). TPA treatment also substantially increased the proportion of cells with predominantly nuclear MRTF-A; this was also ERK-dependent, and sensitive to both the conventional-PKC inhibitor Gö6976 and the pan-PKC inhibitor Gö6983 (*Figure 3B*, *Figure 3—figure supplement 1A*), although only the latter completely inhibited ERK activation and MRTF-A phosphorylation (*Figure 3—figure supplement 1B*). Thus, TPA promotes MRTF-A nuclear accumulation via an ERK- and conventional-PKC- dependent pathway.

While the above observations show an overall positive effect of TPA on MRTF activity, we found that TPA pretreatment of NIH3T3 cells also effectively blocked MRTF-A nuclear accumulation upon subsequent serum stimulation (*Figure 3C*). The inhibitory effect of TPA was insensitive to U0126 and Gö6976, but relieved by Gö6983; the latter effect was independent of ERK, since U0126 had no effect (*Figure 3C*, *Figure 3—figure supplement 1C*). Activation of ERK by the use of a tamoxifen-regulated Raf allele was also unable to inhibit serum-dependent MRTF-A nuclear accumulation (*Figure 3D*). Similar although less pronounced results were observed with transfected MRTF(2–204) PK, in which sequences C-terminal to the RPEL domain were replaced with pyruvate kinase (*Guettler et al., 2008*). In contrast to a previous report (*Muehlich et al., 2008*), we found that serum-induced nuclear accumulation of MRTF-A mutant S544A/T545A/S549A (STS/A) remained normal and was sensitive to TPA pretreatment in our cells (*Figure 3D*; *Figure 3—figure supplement 1D*; see Discussion).

Together these results indicate that TPA also negatively regulates MRTF-A via PKC-dependent signals which act independently of ERK. To characterise this pathway in more detail, we examined the effect of TPA on Rho signalling. Strikingly, TPA treatment inhibited serum-induced F-actin assembly, and this was relieved by Gö6983 but not by U0126 or Gö6976 (*Figure 3E*). TPA also inhibited serum-induced accumulation of Rho.GTP, and phosphorylation of MLC2, independently of ERK but in a PKC-dependent manner (*Figure 3F*, *Figure 3—figure supplement 1E*). Taken together, these results indicate the ability of TPA to activate MRTF-A in NIH3T3 cells represents a balance between a negative effect on Rho-actin signalling, which is mediated by multiple PKC family members, and a positive effect through ERK activation.

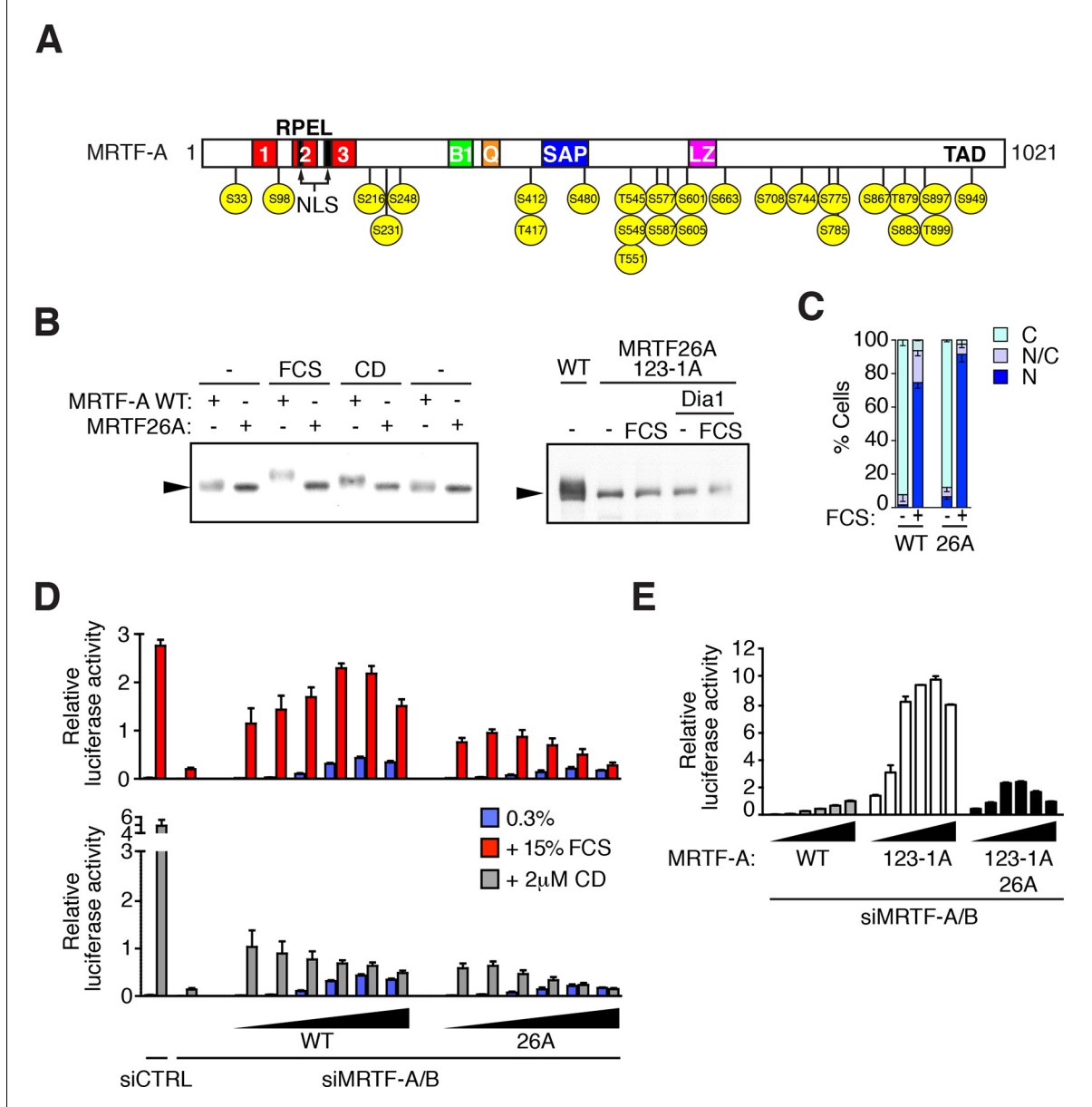

**Figure 2.** MRTF is phosphorylated at multiple sites. (A) Schematic representation of MRTF-A, with S/T-P phosphorylation sites substituted with alanine in MRTF26A indicated in yellow. For details of phosphorylations, see *Table 1* and *Figure 2—figure supplement 1*. (B) Phosphorylation is largely abolished in the MRTF26A mutant. Cells expressing wildtype MRTF-A or MRTF26A (left panel) or MRTF26A-123-1A (right panel), with or without activated mDia1, were stimulated with 15% FCS or 2 μM CD as indicated, and phosphorylation monitored by SDS-PAGE and immunoblotting. See *Figure 2—figure supplement 2*. (C) Cells expressing wild-type MRTF-A or MRTF-A 26A were stimulated with 15% FCS for 30 min. MRTF-A localisation was assessed by immunofluorescence (N, predominantly nuclear; N/C, pancellular; C, predominantly cytoplasmic), in two independent experiments. (D) Cells depleted of endogenous MRTFs were transfected with the MRTF-A responsive 3DA-luc SRF reporter and increasing amounts of wild-type MRTF-A or MRTF-A 26A (3, 6, 12, 25, 50, 100 ng). After treatment with 15% FCS or 2 μM CD, cell extracts were assessed for luciferase activity. Two independent experiments, data presented as mean ± half-range. (E) Cells were transfected with 3DA-luc and MRTF-A derivatives as in (D), and luciferase activity measured 24 hr later. Three independent experiments, data presented as mean ± SEM.

The following figure supplements are available for figure 2:

**Figure supplement 1.** Location of MRTF-A phosphorylation sites.

**Figure supplement 2.** Detection of MRTF-A phosphorylation with phospho-specific antibodies.

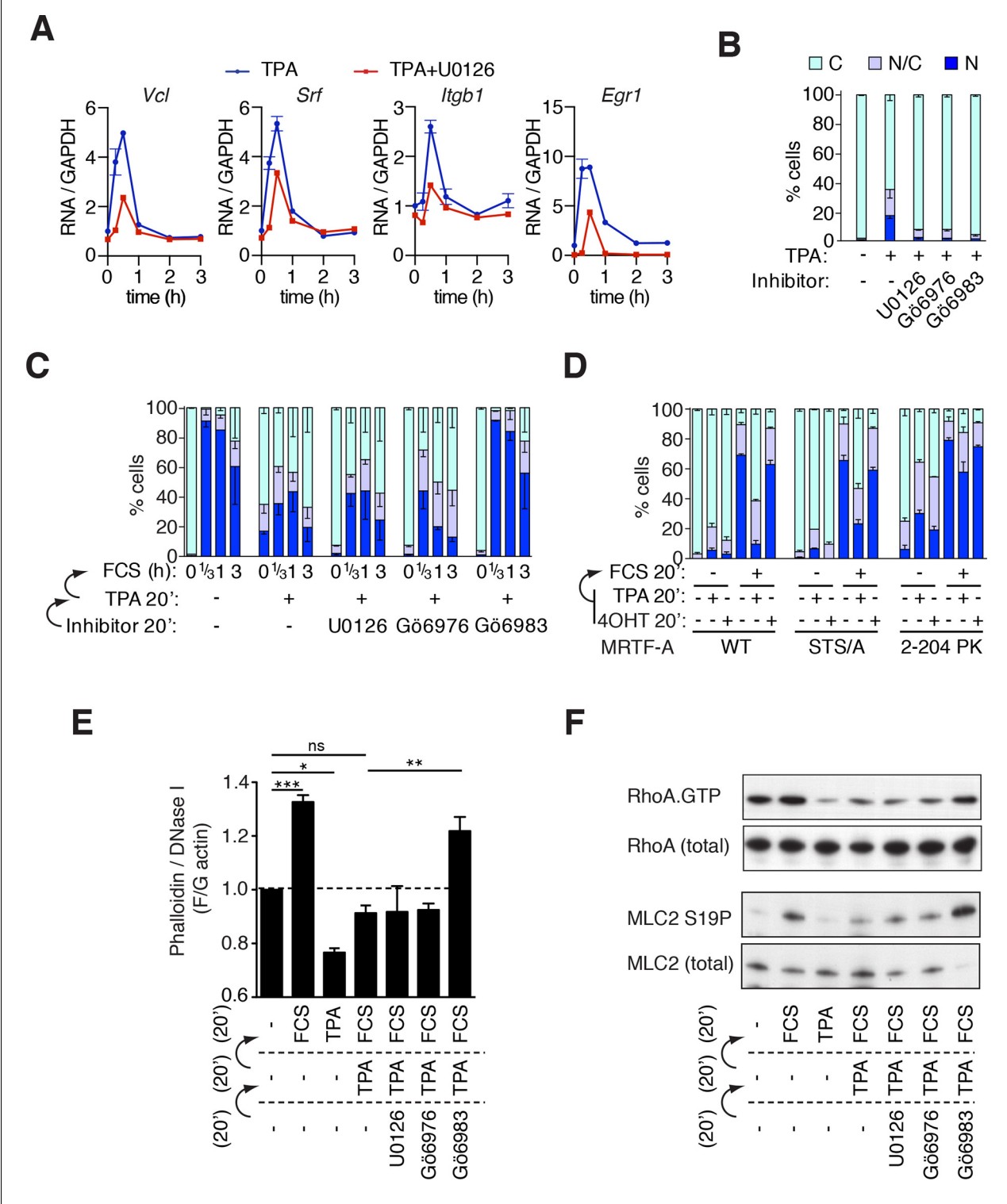

**Figure 3.** Phorbol esters act positively and negatively to control MRTF-A nuclear accumulation. (A) TPA-induction of MRTF target genes. Cells were stimulated with 100 ng/ml TPA, and MRTF-A target gene pre-mRNA at 0.25, 0.5, 1, 2 and 3 hr analysed by qPCR, normalised to GAPDH and expressed relative to uninduced conditions. Data are mean ± half-range, n = 2. (B–D). Cells expressing Flag-MRTF-A or derivatives were pretreated with the indicated inhibitors for 20 min (U0126, 10 μM; Gö6976, 2 μM; Gö6983, 6 μM), then treated with 100 ng/ml TPA for 20', with serum stimulation for 20' as indicated. MRTF-A subcellular localisation was assessed by immunofluorescence. Data are mean ± half-range, n = 2. (B) TPA-induces MRTF-A nuclear accumulation (p<0.01, two-way ANOVA, Bonferroni post-test) but not in presence of inhibitors. (C) Cells as in (B) were stimulated with 15% FCS. Data are mean ± half-range, n = 2. Serum-induced MRTF-A nuclear accumulation was inhibited by pre-treatment with TPA (p<0.01), TPA+U0126 (p<0.01),

*Figure 3 continued on next page*

*Figure 3 continued*

and TPA+Gö6976 (p<0.05), but not TPA+Gö6983 (ns); two-way ANOVA with Bonferroni post-test. (Part B shows only the t = 0 points from this experiment for clarity). (D) Cells expressing Flag-MRTF-A, Flag-MRTF-A STS/A (S544A/T545A/S549A), or MRTF-A(2–204)-PK, together with Raf-ER as indicated, were pre-treated for 20' with 100 ng/ml TPA or 1 μM 4-hydroxytamoxifen (4OHT) then stimulated with 15% FCS for 20 min. Data are from two (WT and STS/A) or four independent experiments ((2–204)-PK). Only pre-treatment with TPA significantly blocked serum-induced nuclear accumulation (Predominantly nuclear MRTF-A cells: WT and STS/A, p<0.01; 2–204 PK, p<0.05; one-way ANOVA, Bonferroni multiple comparison test). (E) TPA inhibition of F-actin assembly in NIH3T3 cells is PKC-dependent but does not require ERK or conventional PKC. Cells were pretreated with the indicated inhibitors for 20 min, treated with 100 ng/ml TPA for 20 min, then stimulated with 15% FCS for 20 min, as in (C). After fixation F- and G-actin were quantified using fluorescently labelled phalloidin and DNase I, respectively, with F:G ratio in unstimulated cells set arbitrarily to unity. Plots are mean ± SEM, n = 3, with significance estimated by one-way ANOVA (*p<0.05; **p<0.01; ***p<0.001). (F) TPA inhibition of serum-induced RhoA activation and MLC2 phosphorylation is PKC-dependent but does not require ERK or conventional PKC. Lysates from cells treated as in panel (E) were used in a RhoA.GTP pull-down assay with GST-rhotekin as the affinity matrix (top); or analysed for MLC2 S19 phosphorylation by immunoblotting (bottom).

The following figure supplement is available for figure 3:

**Figure supplement 1.** Phorbol esters act positively and negatively on MRTF-A nuclear accumulation.

## ERK-mediated S98 phosphorylation and G-actin binding are mutually inhibitory

S98 is located in the 'spacer1' region between RPEL motifs 1 and 2 (*Mouilleron et al., 2011*) and is preceded by a potential 'D-box' type ERK-binding motif (*Tanoue and Nishida, 2003*), which over-laps RPEL1 (*Figure 4A*). We therefore investigated the role of S98 phosphorylation in the assembly of G-actin/RPEL domain complexes (*Figure 4A*). S98 phosphorylation was induced by co-expression of activated Raf or MEK, and by TPA in a U0126-sensitive fashion, indicating that S98 is an ERK target, and MRTF-A phosphorylated at S98 was detectable in the nucleus of stimulated cells (*Figure 4B*; *Figure 4—figure supplement 1A,B*). Serum-stimulated phosphorylation of S98 was impaired by mutations that disrupted the ERK-binding motif and was not observed in the MRTF-A BSAC isoform, which lacks it (*Figure 4C*). Such mutations also impaired the recovery of ERK2 by the MRTF-A N-terminal sequences in pulldown assays (*Figure 4—figure supplement 1C*). We found that G-actin inhibited phosphorylation of S98 by recombinant ERK (*Figure 4D*); this suggests that G-actin must occlude the phosphorylation site and/or compete with ERK docking, although we were unable to observe competition between ERK2 and G-actin binding in pulldown assays (*Figure 4—figure supplement 1D*; see below).

Serine 33 is located at a similar distance N-terminal to the ERK docking motif. It exhibited a high basal level of phosphorylation, which could be slightly increased upon serum or TPA stimulation, and also by expression of activated MEK or Raf (*Figure 4B,E*; *Figure 4—figure supplement 1A*). Induced S33 phosphorylation was U0126-sensitive, indicating the involvement of ERK (*Figure 2—figure supplement 2B,C*) but was not affected by mutation of the ERK-binding motif (*Figure 4E*). Weak ERK docking interaction might explain the weak inducibility of S33 following ERK activation, but it is possible that other proline-directed kinases are responsible for the relatively high S33 phosphorylation in unstimulated cells (*Aoki et al., 2013*).

Previous studies have shown that G-actin forms stable complexes with RPEL1 or RPEL1-spacer1 peptides, and dimeric complexes with spacer1-RPEL2 peptides (*Mouilleron et al., 2011*), but these were unaffected by phosphorylation of S98, or its phosphomimetic substitution with aspartate (*Figure 4—figure supplement 2A,B*). However, in the context of the intact RPEL domain, in which a stable trimeric complex forms on the RPEL1-spacer1-RPEL2 unit (*Mouilleron et al., 2011*), the S98D substitution reduced the stoichiometry of G-actin binding by approximately 1 (*Figure 4F,G*). No further reduction was observed upon simultaneous introduction of the RPEL1 LL74/78AA mutation, which blocks binding of G-actin to RPEL1 (*Figure 4F,G*; *Mouilleron et al., 2008*). Taken together, these data show that phosphorylation of S98 is likely to destabilise the stable trimeric core of the pentameric G-actin/MRTF-A complex (*Mouilleron et al., 2011*), by preventing G-actin association with RPEL1.

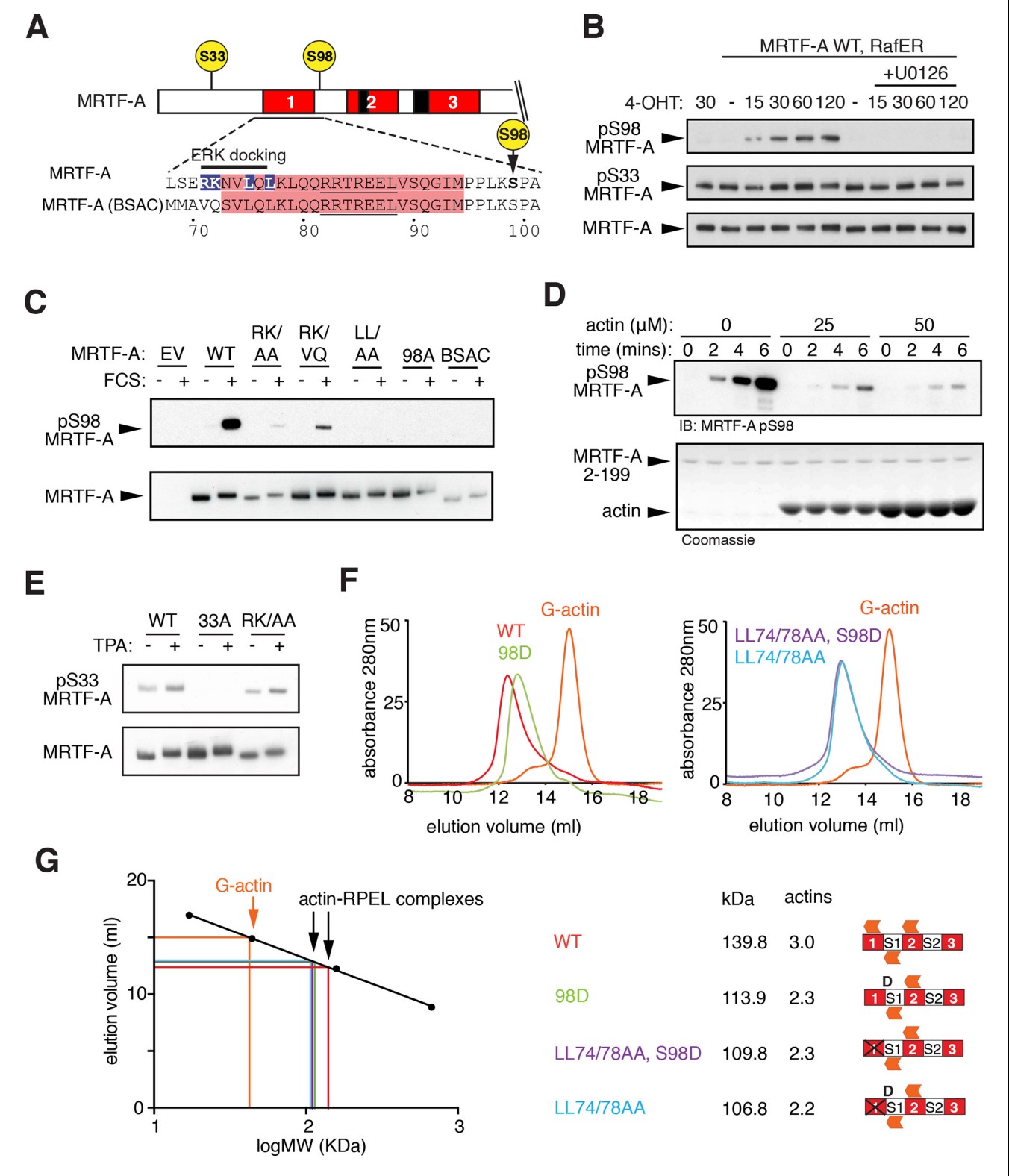

**Figure 4.** ERK-mediated phosphorylation of S98 inhibits actin binding. (**A**) Sequences of MRTF-A(fl) and MRTF-A(BSAC), with RPEL1 highlighted in pink and the ERK-binding motif in blue. (**B**) Cells expressing Flag-MRTF-A and RafER were treated with 4OHT or vehicle and 10 µM U0126 as indicated. Lysates were analysed by immunoblot for MRTF-A pS98, MRTF-A pS33, and MRTF-A (Flag). (**C**) Cells expressing wild-type MRTF-A(fl) or derivatives with the indicated ERK-binding site mutations, or MRTF-A (BSAC), were analysed by immunoblotting with MRTF-A phospho-S98 antibody before and after

*Figure 4 continued on next page*

*Figure 4 continued*

serum stimulation. (**D**) Phosphorylation of recombinant MRTF-A(2-199) by ERK2 in the absence or presence of G-actin was monitored by MRTF-A phospho-S98 antibody (top) or coomassie staining (bottom). (**E**) S33 phosphorylation occurs independently of the ERK-binding motif. Cells expressing MRTF-A derivatives were analysed as in (**C**) using anti-MRTF-A phospho-S33. (**F**) Size exclusion chromatography of complexes formed between G-actin and recombinant MRTF-A(67–199) derivatives, or G-actin alone, colour coded. (**G**) Left, calibration curve; right, apparent stoichiometry of G-actin binding.

The following figure supplements are available for figure 4:

**Figure supplement 1.** ERK phosphorylation of MRTF-A S98.

**Figure supplement 2.** S98 phosphorylation impairs assembly of MRTF-A / G-actin complexes.

## MRTF-A N-terminal phosphorylations affect nucleocytoplasmic shuttling

Although the MRTF-26A mutation did not affect subcellular localisation of MRTF-A in resting cells or its signal-induced nuclear accumulation, the effects of S98 phosphorylation on G-actin binding prompted further investigation of the role of MRTF-A N-terminal phosphorylation on nucleocytoplasmic shuttling. To simplify the analysis given the numerous phosphorylation sites elsewhere in the molecule, we first investigated their role in regulation of MRTF-A(2–204)-PK, whose regulation is similar to MRTF-A itself (*Vartiainen et al., 2007*). TPA stimulation or co-expression of activated MEK promoted nuclear accumulation of MRTF-A(2–204)-PK (*Figure 5A*). In contrast, the S98A derivative of MRTF-A(2–204)-PK was refractory to TPA stimulation but remained slightly inducible by MEK, possibly reflecting indirect effects of chronic ERK activation, while the S98D phosphomimetic derivative was significantly nuclear in unstimulated cells and unaffected by TPA or MEK (*Figure 5A*). Similar results were seen following short-term activation of Raf (*Figure 5B*). G-actin binding occludes the importin αβ-dependent NLS within the RPEL domain (*Hirano and Matsuura, 2011*; *Mouilleron et al., 2011*; *Pawłowski et al., 2010*). Our observations therefore suggest a model in which S98 phosphorylation promotes MRTF nuclear import indirectly, by inhibiting G-actin binding and thereby facilitating recruitment of importin αβ to the NLS.

We also used MRTF-A(2–204)-PK to investigate the role of S33 phosphorylation. To our surprise, alanine substitution at S33 significantly increased MRTF-A(2-204)-PK nuclear accumulation in resting cells; while combination of S33A with S98A had no significant effect, its combination with S98D resulted in predominantly nuclear accumulation of the protein (*Figure 5B*). In contrast, aspartate substitution of S33 reduced basal nuclear accumulation of MRTF-A(2–204)-PK; there was little response to short-term activation of ERK by RafER, and combination with either S98 mutation had no further significant effect (*Figure 5B*). These data suggest that in contrast to S98 phosphorylation, S33 phosphorylation promotes cytoplasmic localisation of MRTF-A.

Next, we studied the effects of S98 and S33 mutations on regulation of the full-length protein. As mentioned above, MRTF-A exhibits ERK-dependent nuclear accumulation in response to TPA stimulation (*Figure 3B*, *Figure 3—figure supplement 1A*). In agreement with the results with MRTF-A(2–204)-PK, alanine substitution of S98 significantly inhibited, but did not block, TPA-induced nuclear accumulation, whereas alanine substitution of S33 increased basal levels of nuclear MRTF-A (*Figure 5C*). Although aspartate substitution of S98 slightly increased basal levels of nuclear MRTF-A, it also partly inhibited its TPA-induced nuclear accumulation. Finally, aspartate substitution of S33 blunted the TPA induced nuclear accumulation of full length MRTF-A (*Figure 5C*). Taken together, these results suggest that the S98 and S33 phosphorylations contribute in opposing ways to the regulation of MRTF-A. We will return to this issue below.

## A Crm1-dependent NES is located N-terminal to the RPEL domain

The finding that S33 phosphorylation inhibits MRTF-A(2–204)-PK nuclear accumulation led us to investigate whether it affects nuclear import or export. MRTF-A localisation is regulated primarily at the level of Crm1-mediated nuclear export, which is facilitated by G-actin/MRTF interaction (*Vartiainen et al., 2007*). To map Crm1-dependent export activity in MRTF-A N-terminal sequences, we transfected NIH3T3 cells with derivatives of RevΔGFP, an NES-deleted HIV Rev protein, which is

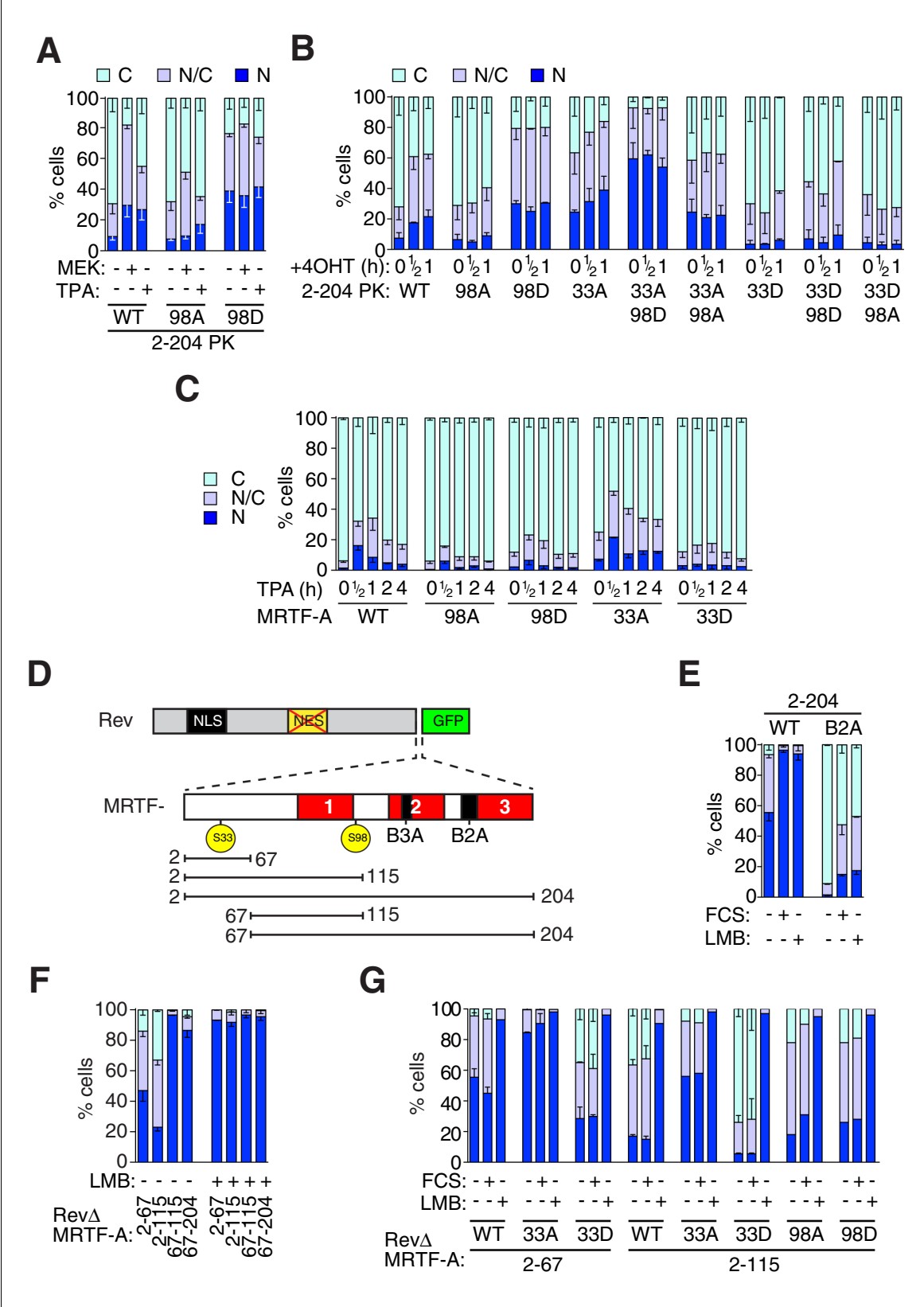

**Figure 5.** Phosphorylation of S98 and S33 has opposing effects on MRTF-A nuclear accumulation. (**A**,**B**) Cells expressing MRTF-A derivatives, with or without activated MEK or RafER, were treated as indicated, and their subcellular localisation scored by immunofluorescence. (**A**) ERK-mediated S98
*Figure 5 continued on next page*

Figure 5 continued

phosphorylation controls MRTF-A(2–204)PK localisation. Data are mean ± SEM, n = 3. Mutant S98D baseline higher than wildtype or S98A (p<0.01); MEK and TPA potentiate wildtype (p<0.001); 98A increased by MEK (p<0.01; possibly reflecting chronic ERK stimulation); others not significant. Two-way ANOVA with Bonferroni post-test. (B) Alanine substitutions or 'phosphomimetic' aspartate substitutions at S33 and S98 have opposing effects on MRTF-A(2–204)PK subcellular localisation. Data are mean ± half-range, n = 2. Mutant S98D baseline higher than wildtype or S98A (p<0.01); S98D and S33A cooperate to elevate baseline (p<0.001); 4OHT significantly induces wildtype and S33A (both p<0.05); two-way ANOVA with Bonferroni post-test. (C) N-terminal phosphosite mutations affect nucleocytoplasmic shuttling of intact MRTF-A. Data are mean ± half-range, n = 2. TPA treatment significantly increases nuclear localisation of wild-type MRTF-A (p<0.001), 33A (p<0.001) and 98D (p<0.05); one-way ANOVA, Bonferroni multiple comparison test. (D) Schematic representation of the MRTF-A / RevΔGFP derivatives. (E–G) analysis of MRTF-A NES activity by RevΔGFP assay. (E) MRTF-A(2–204) contains a Crm1-dependent NES. Data are mean ± half-range, n = 2. (F) MRTF-A residues 2–67, but not the RPEL domain, function as an NES. Data are mean ± SEM, n = 3. The 2–67 and 2–115 derivatives differ significantly from each other and from 67–115 and 67–204 in resting cells (all p<0.001 except 2–67 vs 67–204, p<0.01), but not following LMB treatment; two-way ANOVA with Bonferroni post-test. (G) Alanine or 'phosphomimetic' aspartate substitutions at S33, but not S98 affect MRTF-A N-terminal NES activity. Data are mean ± half-range, n = 2; absence of error bar shows a single datapoint (ie mean of two technical replicates).

The following figure supplement is available for figure 5:

**Figure supplement 1.** Use of RevΔGFP to detect NES activity in MRTF-A.

completely nuclear in unstimulated cells, but whose cytoplasmic localisation can be restored upon insertion of NES elements (*Figure 5D*, *Figure 5—figure supplement 1A*, *Henderson and Eleftheriou, 2000*). Insertion of the MRTF-A N-terminal sequences and RPEL domain into RevΔGFP rendered it partially cytoplasmic, and this was dependent on the integrity of the G-actin -binding sites (*Figure 5E*, *Figure 5—figure supplement 1B*). Nuclear localisation could be restored by serum stimulation, or by treatment with Leptomycin B (LMB), indicating Crm1 involvement in nuclear export (*Figure 5E*). The RPEL domain contains the major MRTF-A NLS (*Hirano and Matsuura, 2011*; *Pawłowski et al., 2010*), and inactivation of this sequence substantially increased cytoplasmic localisation of RevΔMRTF-A(2–204) in unstimulated cells (*Figure 5E*, *Figure 5—figure supplement 1B*). The N-terminal sequences of MRTF-A, therefore, contain a Crm1-dependent nuclear export signal.

To map the NES, we tested different fragments of the MRTF-A N-terminal sequences in the RevΔ assay. The sequences flanking the RPEL domain, MRTF-A(2–67), were sufficient to promote Crm1-dependent nuclear exclusion of RevΔ, and the efficiency of this was increased when these were extended to include the RPEL1 and spacer1 (*Figure 5F*); although no regulation in response to serum stimulation was observed both proteins accumulated in the nucleus upon LMB treatment (*Figure 5G*). Substitutions at the S33 phosphoacceptor affected localisation even in constructs lacking the intact MRTF RPEL domain, with S33A inhibiting export in unstimulated cells and S33D promoting it (*Figure 5G*; *Figure 5—figure supplement 1C*). In contrast, neither the RPEL domain itself, nor the RPEL1 and spacer1 sequences, promoted cytoplasmic localisation of RevΔ (*Figure 5F*).

The above results show that S33 phosphorylation must specifically affect export. An alanine-triplet scan across residues 2–67 revealed that NES activity in the MRTF-A N-terminal sequences was strongly dependent on residues 20–22 (DDE) and 41–49 (NELQELSLQ) which conforms to the classical Crm1-binding site consensus (*Güttler and Görlich, 2011*; *Henderson and Eleftheriou, 2000*). Alanine substitution of the 'anchor' (ΦΧΦ) residues 46–48 also resulted in cytoplasmic localisation of RevΔ MRTF derivatives (*Figure 6A*, *Figure 6B*; see Discussion). Consistent with this, MRTF residues 2–67 could recover recombinant Crm1 from solution in a pulldown assay, and this required both Ran.GTP and the integrity of the anchor residues (*Figure 6C*).

## The N-terminal NES cooperates with multiple MRTF NES elements

We next sought to relate the activity of the N-terminal NES to regulation of intact MRTF-A. Previous studies have mapped a Crm1-dependent NES in the Q-rich region (NES2 in *Figure 7A*, *Hayashi and Morita, 2013*; *Muehlich et al., 2008*). However, MRTF-A derivatives containing NES2 but lacking C-terminal sequences are constitutively nuclear, and MRTF-A N-terminal fusion proteins that lack NES2 remain regulated (*Guettler et al., 2008*; *Miralles et al., 2003*), suggesting that further regulatory sequences must exist. We used the online tools NetNES, ValidNES and NESmapper (*Fu et al., 2013*; *Kosugi et al., 2014*; *la Cour et al., 2004*, and K.Kirli and D. Gorlich, personal communication)

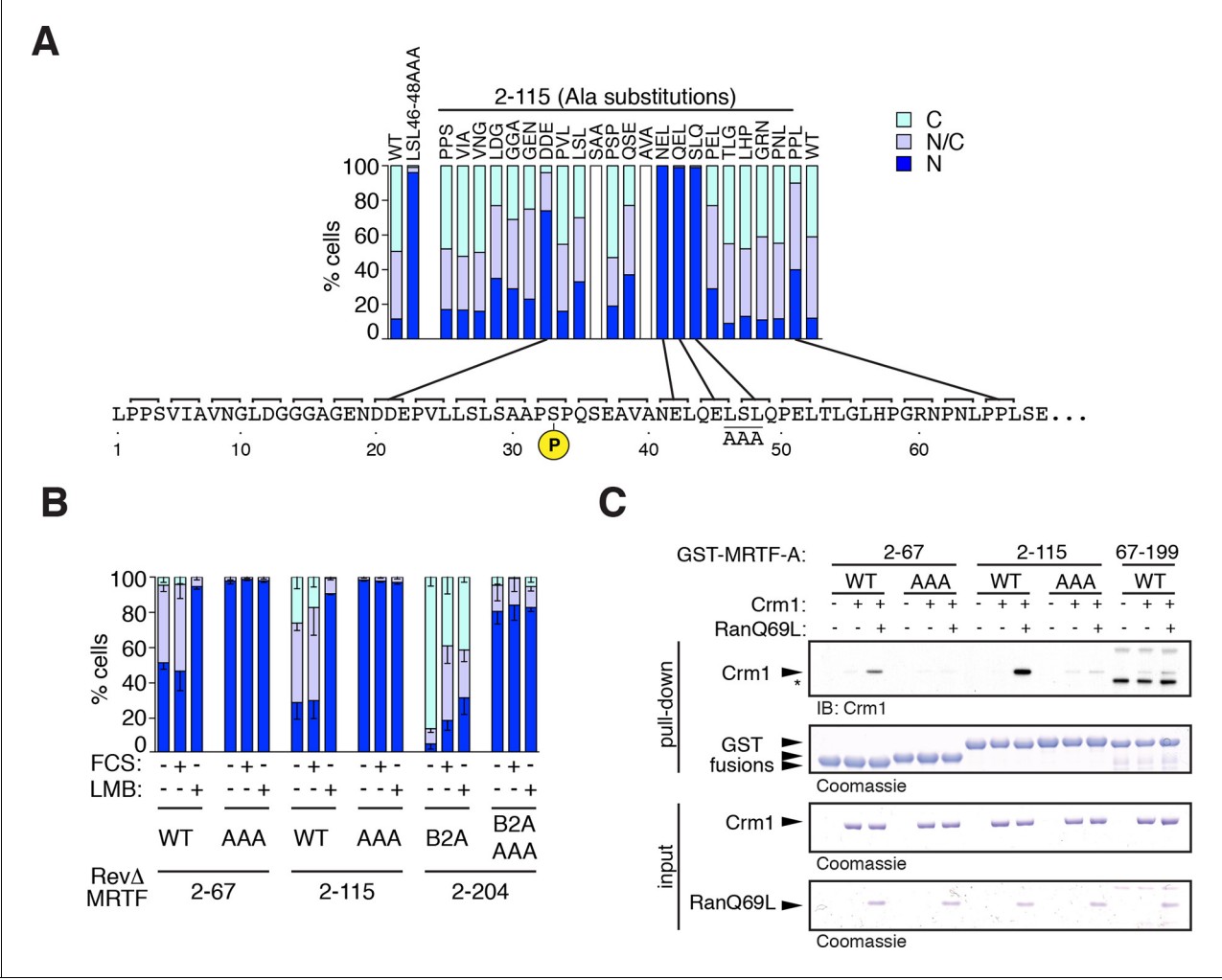

**Figure 6.** Mapping the Crm1-dependent N-terminal NES. Subcellular localisation of RevΔGFP derivatives was scored by immunofluorescence. (**A**) Alanine-scan derivatives of RevΔ(2–115) GFP with the S33 phosphoacceptor and the LSL46-48AAA NES anchor-site mutations indicated. A representative experiment is shown. (**B**) Mutation of the NES core inactivates it. Data are mean ± SEM, n = 3. 2–67, 2–115, B2A(2–204) are significantly different from each other (all p<0.001); only B2A (2–204) exhibits significant change upon FCS stimulation (p<0.001); AAA mutants are significantly more nuclear than their parent (p<0.001); 2-way ANOVA with Bonferroni post-hoc test; 2–67/AAA comparison for N cells only). (**C**) The NES core is required for Ran-dependent Crm1 binding in vitro. Recombinant GST MRTF-A derivatives, with and without the NES anchor mutation, were used in pulldown assays with recombinant Crm1 and Ran 1–180 Q69L.

to identify 10 potential NES elements (a–j) within MRTF-A (*Figure 7A*; *Figure 7—figure supplement 1A*). Four sequences were not studied further: one, within the LZ element, was previously found to lack activity (*Figure 7—figure supplement 1A* element (g), *Muehlich et al., 2008*); two others, within the N-terminal sequences (*Figure 7—figure supplement 1A* elements (b) and (c), K.Kirli and D.Gorlich, personal communication, *Hayashi and Morita, 2013*) did not score in the RevΔ assay (*Figure 7—figure supplement 1B,C*), while the fourth scored only with one algorithm (*Figure 7—figure supplement 1A*, element (h)).

We tested whether deletions encompassing the remaining six potential NES sites, denoted NES1-6 (*Figure 7A*, *Figure 7—figure supplement 1A*), affected the localisation of intact MRTF-A in unstimulated cells, singly or in combination. A short deletion encompassing site NES2, within the Q-rich element, increased nuclear accumulation of MRTF-A in unstimulated cells, as seen previously (*Muehlich et al., 2008*), as did deletion of NES5/6, while the deletion of the other two sites alone had no appreciable effect (*Figure 7B*). Introduction of the N-terminal NES1 anchor mutation into

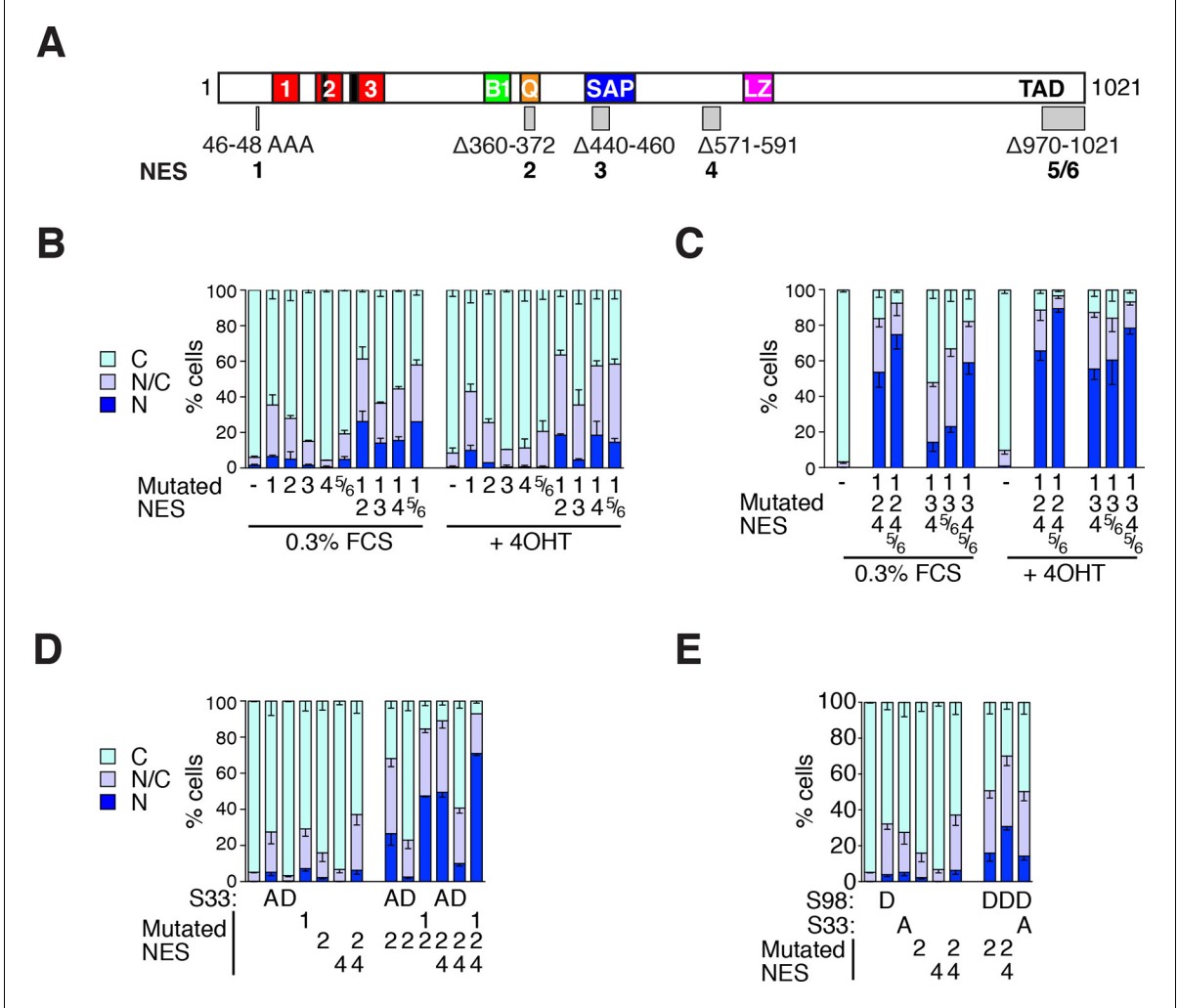

**Figure 7.** MRTF-A contains multiple cooperating NES elements. Cells expressing MRTF-A derivatives, with or without RafER, were treated as indicated and their subcellular localisation was scored by immunofluorescence. (A) Summary of MRTF-A NES predictions and NES-inactivating deletions and point mutations (see also *Figure 7—figure supplement 1A*). (B–E) MRTF-A NES elements functionally cooperate with each other (B,C) and with N-terminal phosphorylations (D,E) to maintain cytoplasmic localisation of full-length MRTF-A. Data in panels D and E are from the same experiment. Data are mean ± half-range n = 2 (B,D,E) or mean ± SEM, n =3 (C). (B) NES1 cooperates with the C-terminal NES elements. Nuclear accumulation is enhanced by inactivation of NES1 ($p < 0.001$), NES2 ($p < 0.001$) or NES5/6 ($p < 0.05$); inactivation of NES1 enhances the effect of inactivating NES2 and NES5/6 ($p < 0.001$); one-way ANOVA with Bonferroni multiple comparison test. (C) Inactivation of NES5/6 significantly enhances effect of NES1/2/4 inactivation ($p < 0.001$); one-way ANOVA with Bonferroni multiple comparison test. (D) Effect of alanine substitution at S33 is similar to NES1 inactivation. S33A enhances the effect of inactivation of NES2 ($p < 0.05$) or NES2+4 ($p < 0.001$), and is not significantly different from NES1 inactivation; NES1 inactivation enhances inactivation of NES2+NES4 ($p < 0.001$); one-way ANOVA with Bonferroni multiple comparison test. (E) The S98D phosphomimetic mutation cooperates with inactivation of NES2 ($p < 0.01$) and NES2+4 ($p < 0.001$). One-way ANOVA, Bonferroni's multiple comparison test.

The following figure supplement is available for figure 7:

**Figure supplement 1.** NES predictions and analysis.

each of these MRTF-A derivatives resulted in its nuclear accumulation in a substantially greater proportion of resting cells compared with the wildtype protein (*Figure 7B*). The various NES mutants showed no change in subcellular localisation upon activation of ERK (*Figure 7B*). Combination of multiple sites further increased the proportion of cells exhibiting nuclear MRTF-A (*Figure 7C*).

Finally, we tested whether S33 and S98 phosphorylation, which promote nuclear export and import respectively, functionally cooperate with the NES elements. Alanine substitution of S33, which inhibits the activity of NES1, enhanced MRTF-A nuclear accumulation brought about by inactivating mutations of NES2, or NES2 and NES4, to a degree only slightly less than alanine substitution of the NES1 core; aspartate substitution of S98, which inhibits actin binding, substantially enhanced the effect of inactivating mutations at NES2, or NES2 and NES4 (*Figure 7D,E*). Taken together, these data show that MRTF-A contains multiple Crm1 dependent NES elements located throughout its length, which function cooperatively with each other, and with the N-terminal phosphorylation sites at S33 and S98, to control MRTF-A subcellular localisation in resting cells.

## Discussion

We characterised the role of phosphorylation and nuclear export in the regulation of mouse MRTF-A, a member of the myocardin family of SRF transcriptional coactivators. We found that G-actin binding exerts a repressive role on MRTF-A phosphorylation, which acts both positively and negatively in MRTF-A regulation in fibroblasts. Multiple phosphorylations throughout MRTF-A potentiate transcriptional activation. In addition, phosphorylation at S98, within the RPEL domain, inhibits G-actin binding and promotes nuclear accumulation, while phosphorylation of S33 N-terminal to the RPEL domain promotes nuclear export. We identified five new MRTF-A NES elements, one of which is controlled by S33 phosphorylation, and found that these elements, and S98 phosphorylation status contribute to the maintainance of cytoplasmic MRTF-A in resting cells (*Figure 8A*).

### G-actin binding controls MRTF-A phosphorylation

We previously demonstrated that both Rho and ERK signalling contribute to MRTF-A phosphorylation (*Miralles et al., 2003*). Three lines of evidence indicate Rho-mediated MRTF phosphorylation results from the dissociation of G-actin-MRTF complexes following depletion of the G-actin pool: it is induced by (i) active forms of the Rho effector mDia, which promote actin polymerization; (ii) mutation of the RPEL motifs, which render MRTF-A unable to bind actin; and (iii) treatment with cytochalasin D, which prevents G-actin from binding to MRTF-A. These treatments also promote MRTF nuclear accumulation, but relocalisation of MRTF-A to the nucleus by inactivation of Crm1, or by fusion to an active nuclear import signal, does not induce phosphorylation unless actin association is also disrupted. Our data thus support a model in which the interaction of G-actin with MRTF-A either inhibits access of kinases to MRTF or facilitates its interaction with nuclear phosphatase(s) (*Figure 8A*). Consistent with this, although myocardin, which does not bind G-actin and is constitutively nuclear, is constitutively phosphorylated, exchange of its RPEL domain for that of MRTF-A is sufficient to confer actin-regulated phosphorylation. There is significant interaction between MRTF-A and G-actin in the nucleus, and our experiments with NLS fusions and Crm1 inhibitors show that interaction between nuclear G-actin and MRTF-A must contribute to regulation of its phosphorylation.

### Multisite MRTF-A phosphorylation contributes to transcriptional activation

We identified 35 phosphorylations on MRTF-A in serum-stimulated cells, of which 24 are associated with S/T-P motifs, the core recognition sequences for MAP- and cyclin-dependent kinases. Consistent with this, MRTF phosphorylation is sensitive to MEK inhibition, and is potentiated by TPA and Raf activation. Most MRTF-A sites tested were phosphorylated to some extent in resting cells, consistent with analyses of unstimulated cells in the PHOSIDA and phosphosite databases (http://www.phosida.com, http://www.phosphosite.org/, *Gnad et al., 2011*). Sensitivity of much MRTF-A phosphorylation to U0126 suggests that many of the sites, especially those containing S/T-P motifs must be targets for ERK phosphorylation, consistent with recent studies showing that ERK exhibits stochastic activation in unstimulated cells (*Aoki et al., 2013*). Many of the same MRTF-A S/T-P sites become phosphorylated upon actin dissociation, although the identity of the kinase(s) involved remains unclear.

We constructed an MRTF-A mutant, 26A, in which all high-confidence S/T-P phosphoacceptor sites were substituted with alanine. This mutant exhibited no obvious abnormalities in nucleocytoplasmic shuttling but was impaired in its ability to activate SRF-controlled reporter genes. A similar

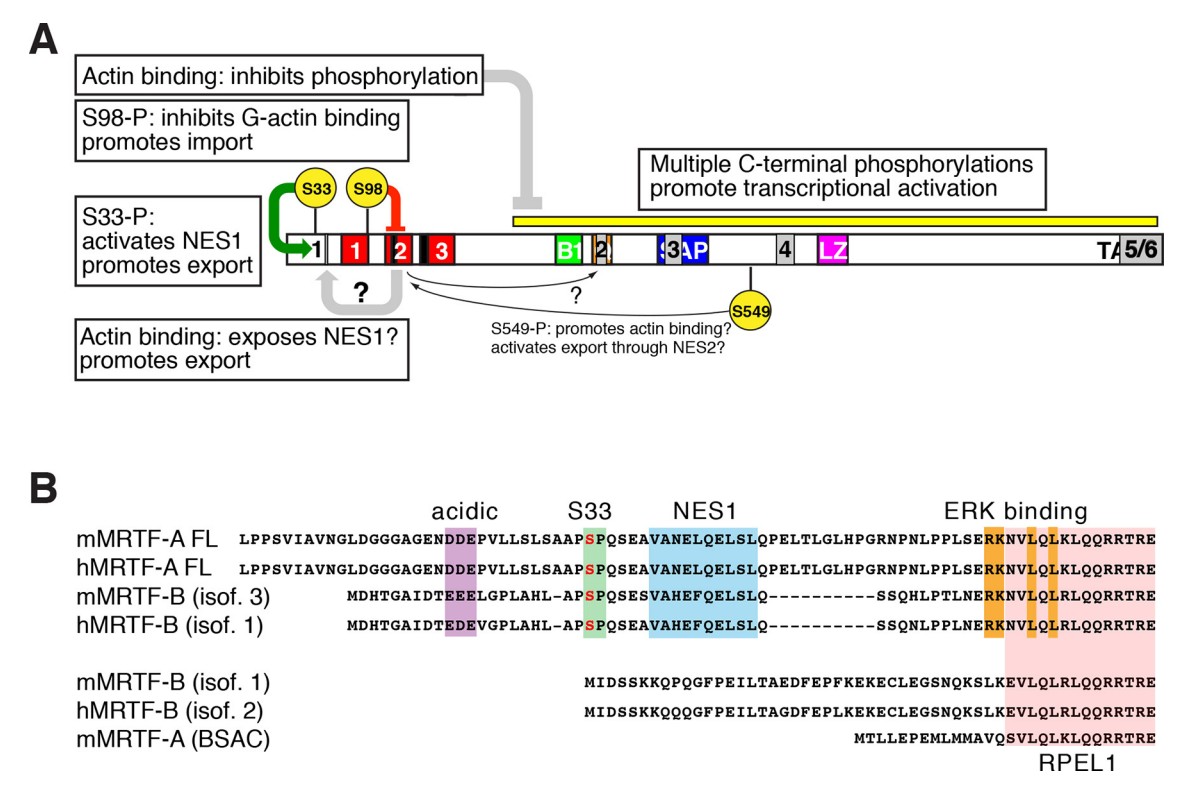

**Figure 8.** Roles of phosphorylation, NES elements, and actin binding in MRTF-A nucleocytoplasmic shuttling. (**A**) Positive and negative roles of phosphorylation in MRTF-A, shown as in (*Figure 7A*) with the NES elements superposed in grey. G-actin binding inhibits MRTF-A phosphorylation, which is required for effective transcriptional activation. S98 phosphorylation inhibits formation of multimeric G-actin/MRTF complexes on the RPEL domain, promoting MRTF-A nuclear import; S33 phosphorylation potentiates the activity of NES1, which may be occluded in the absence of actin binding to the RPEL domain. The regulatory role of S549 phosphorylation in G-actin binding and NES2 activity (*Muehlich et al., 2008*), is indicated but is not detectable in our system. (**B**) Both MRTF-A and MRTF-B express variants containing or lacking the NES1 element and ERK docking site (Genbank: mouse MRTF-A FL, NP 694629.2 and AF532597.1; MRTF-A BSAC AAM 94258.1; MRTF-B isoform 3, NP 001116139.1; isoform 1, NP 705816.2, AAN 33042.1).

defect was apparent when the 26A mutation was introduced into the MRTF-A derivative MRTF-A-123A, which cannot bind actin and is constitutively nuclear (*Guettler et al., 2008*). These results strongly suggest that phosphorylation of MRTF contributes positively to transcriptional activation, but more work is necessary to identify the specific sites involved and how their phosphorylation affects operation of the transcriptional machinery. In contrast, previous studies have suggested a negative regulatory role for GSK3-mediated phosphorylation of myocardin at residues S455/S459/S463/S467 (*Badorff et al., 2005*), conserved across the Myocardin family, possibly by controlling its degradation (*Charbonney et al., 2011*; *Xie et al., 2009*). Further work is required to resolve this issue.

## S98 phosphorylation and actin binding are mutually antagonistic

Within the RPEL domain, S98 is phosphorylated by ERK upon serum stimulation, and exhibits very low basal phosphorylation in unstimulated cells; the corresponding MRTF-B residue is also phosphorylated (*Olsen et al., 2006*). S98 phosphorylation also requires an ERK docking motif which overlaps the RPEL domain. Biochemical studies show that S98 phosphorylation and G-actin binding are mutually antagonistic, and that S98 phosphorylation promotes the nuclear accumulation of MRTF-A. This could contribute to MRTF regulation in two ways. First, in unstimulated cells high basal levels of S98 phosphorylation might potentiate nuclear accumulation of MRTF-A, adjusting expression of MRTF-SRF target genes accordingly. Second, rapid phosphorylation of S98 following signal-induced

G-actin depletion might delay G-actin rebinding, thereby prolonging the effective period of MRTF-A activation, in a manner analogous to the phosphorylation of WASP contingent upon GTPase activation (*Torres and Rosen, 2003*). In resting cells, the MRTFs interact with G-actin in both compartments (*Baarlink et al., 2013*; *Vartiainen et al., 2007*). Our finding that S98-phosphorylated MRTF-A is mostly nuclear following ERK activation suggests that at least in this context, it must affect nuclear G-actin-MRTF-A interactions (*Vartiainen et al., 2007*).

## Multiple NES elements cooperate with phosphorylation to control MRTF-A nuclear accumulation

We identified five new MRTF-A NES elements. These elements functionally cooperate with each other, and with phosphorylation of S98, to promote the cytoplasmic localisation of MRTF-A in resting cells (*Hayashi and Morita, 2013*; *Muehlich et al., 2008*). All except NES4 are conserved between MRTF-A and MRTF-B, and through evolution, although NES3 also overlaps the SAP domain (*Figure 8A*). NES1 activity is potentiated by phosphorylation or acidic substitution of nearby S33 (*Figure 8B*). It is likely that this may facilitate Crm1 interaction, since Crm1-dependent NES elements show a strong preference for negatively charged residues at positions 10–15 and 24–28 N-terminal to the hydrophobic anchor (*la Cour et al., 2004*). Phosphorylation of S33 was readily detectable in resting cells, and enhanced only slightly by serum stimulation, suggesting that it controls basal activity of NES1 rather than signal-regulated MRTF-A nuclear accumulation. Together, the opposing roles of S33 and S98 phosphorylation potentially allow fine tuning both of the basal levels of nuclear MRTF-A in resting cells, and the kinetics with which the system can return to its initial state following signalling. MRTF-A and MRTF-B express tissue-specific evolutionarily conserved N-terminal isoforms (*Ishikawa et al., 2013*), that contain both NES1 and the ERK docking sites required for effective S98 phosphorylation (*Figure 8B*). This suggests that coordination of MRTF NES1 activity and S98 phosphorylation is biologically significant, but the biological context for this remains to be determined. Unlike NES1, the other NES elements were not closely associated with phosphorylation sites.

Phosphorylation of S549, and possibly S544 and T545, overlapping the GSK3 site, was previously reported to negatively regulate MRTF-A by promoting G-actin binding to the RPEL domain and NES2-mediated nuclear export (*Muehlich et al., 2008*). We found that alanine substitution of these residues did not affect MRTF-A nuclear localisation in our system. Phosphorylation of S549 cannot be strictly essential for the RPEL domain itself to display G-actin-regulated shuttling properties, since MRTF-A N-terminal fusion proteins which lack it remain regulated (*Guettler et al., 2008*), and it is conserved in myocardin, which is nuclear and does not shuttle (*Guettler et al., 2008*; *Wang et al., 2001*). Further work is clearly required to clarify the apparently complex interplay between phosphorylation by ERK and GSK3, G-actin binding, and MRTF-A nucleocytoplasmic shuttling. In related experiments, we found that TPA pretreatment inhibits serum-induced MRTF-A nuclear accumulation through down-regulation of RhoA rather and was independent of ERK.

Nuclear export of MRTF-A is dependent on the integrity of the G-actin-binding sites within the RPEL domain (*Vartiainen et al., 2007*). This suggests that nuclear actin functionally cooperates with Crm1 in MRTF-A export (*Figure 8A*), although we cannot rule out the possibility that it reflects competition between cytoplasmic G-actin and importin αβ for binding to MRTF-A once the Crm1-MRTF complex reaches the cytoplasmic surface of the nuclear pore. It is unlikely that there is direct interaction between Crm1 and the RPEL domain, since the isolated MRTF-A RPEL domain functions neither in the RevΔ assay nor when substituted for the Phactr1 RPEL domain, which is regulated independently of Crm1 (*Wiezlak et al., 2012*). Moreover, our experiments show that the isolated N-terminal NES1 can function in the absence of actin binding, even though fusion proteins containing it and the RPEL domain require actin binding for export. We speculate that NES1 may somehow be occluded by the RPEL domain in the absence of bound G-actin (*Figure 8A*). Interestingly, a recent report implicated the Ddx19 helicase in conformation-dependent control of MRTF-A nucleocytoplasmic shuttling (*Rajakylä et al., 2015*). It will be interesting to investigate the interplay between phosphorylation, G-actin binding and MRTF nucleocytoplasmic shuttling.

# Materials and methods

## Plasmids

Expression plasmids: MRTF-A derivatives were generated from pEF-Flag-MALfl-sir, which expresses mouse MRTF-A (fl) (*Miralles et al., 2003*) but includes the sequence change 5'-521AGCTGGTG-GAGA532-3' to 5'-521A<u>A</u>CT<u>A</u>GT<u>A</u>GA<u>A</u>A532-3' to render it resistant to our MRTF-A/B siRNA. MRTF-123-1A and B2A are as described (*Pawłowski et al., 2010*; *Vartiainen et al., 2007*), other derivatives are as described in the text. Other plasmids were Flag-MRTF-A(BSAC)(*Miralles et al., 2003*); Myocardin (MC), MC-N12-MRTF-A, MRTF-A(2–204)PK (*Guettler et al., 2008*); pRev(1.4)-GFP and pRev(1.4)-NES-GFP (gift from *Henderson and Eleftheriou, 2000*) (MRTF-A sequences inserted as BamHI-AgeI fragments); constitutively active mDia1 FH1/FH2 (*Copeland and Treisman, 2002*); pH10zz-[TEV]-MmCrm1 and pH10zz-[TEV]-HsRan(1–180) Q69L (*Güttler et al., 2010*); pBABE-MEK-R4F (MEK DN3/218E/222D, *Mansour et al., 1994*) and pBABE-RafER were from Axel Behrens. GST fusion proteins were expressed using pET-41a-3CΔ, which has a 3C-protease site substituted for the enterokinase site, and all restriction sites 5' to BamHI deleted.

## Proteins and peptides

Peptides were synthesised by the London Research Institute (LRI) peptide synthesis platform. MRTF-A(94–142), FAM-MRTF-A(62–104) and FAM-MRTF-A(67–98) derivatives are as in *Figure 4—figure supplement 2A*. Phosphorylated 10–12 mer peptides for immunizations were centred on the phosphoacceptor site (see *Table 1*). Actin was prepared as described (*Mouilleron et al., 2008*).

GST-fusions were expressed in *E. coli* Rosetta (DE3) pLysS (Novagen). Lysis was in 50 mM Tris-HCl pH 7.5, 300 mM NaCl, 1% TX-100, 5 mM DTT, 10 mM EDTA pH 8, 1 mM PMSF, 15 µg/mL Benzamidine, followed by passage through a French press. Following adsorption onto glutathione-Sepharose, proteins were recovered by cleavage with GST-3C protease at 4°C overnight in 50 mM Tris-HCl (pH 7.5), 100 mM NaCl, 1 mM DTT. Crm1 and Ran(1–180) Q69L were purified as described (*Güttler et al., 2010*).

For GST pulldown assays, glutathione-sepharose was saturated with GST-fusion proteins, washed in binding buffer (50 mM Tris-HCl pH 7.5, 100 mM NaCl, 10 mM MgCl$_2$, 0.05% NP-40), and incubated with purified recombinant Crm1, Ran(1–180) Q69L or ERK2 for 3 hr at 4°C. After four washes with binding buffer, proteins were eluted in SDS loading buffer for analysis. For kinase assays, glutathione-Sepharose was saturated with GST-MRTF 2–199, washed in binding buffer, and incubated with purified recombinant ERK2 and actin in binding buffer with 250 µM ATP and 10 mg/ml BSA, at 30°C. The reaction was stopped by addition of SDS loading buffer.

RhoGTP pull-downs used Rho activation assay (Millipore) with modifications. Cells were grown in 15 cm dishes, washed twice in ice-cold TBS, scraped in 400 µl 2x lysis buffer with 16% glycerol, made to 800 µl with water, and clarified. 20 µl was retained as input, and the remainder incubated with 20 µl GST-Rhotekin for 45 min at 4°C. After three 1 ml washes with Lysis buffer, proteins were eluted with SDS loading buffer and Rho recovery assessed by immunoblotting.

## Transfections and gene expression assays

NIH3T3 cells were cultured in DMEM, 10% FCS. For stimulation experiments cells were maintained in 0.3% FCS for 20 hr, then stimulated with 15% FCS, 2 µM CD, or 100 ng/ml TPA, with 10 µM U0126 (sufficient to inhibit MEK activation without substantial effects on other kinases, *Bain et al., 2007*), 2 µM Gö6976 or 6 µM Gö6983 (Gö6976 concentration insufficient to block non-novel PKC isoforms, *Martiny-Baron et al., 1993*), 1 µM LatB or 50 nM LMB as required. For mass spectrometry, a tetracycline-inducible cell line expressing MRTF-A(fl)sir was constructed, maintained in 0.3% FCS, and induced with tetracycline for 48 hr before FCS stimulation.

For immunofluorescence (*Vartiainen et al., 2007*), NIH3T3 cells were transfected in six-well plates (150000 /well), with 50 ng pEF-Flag-MRTF-PK derivatives, 100 ng pEF-MRTF-A derivatives, 50 ng MEK-R4F, 100 ng RafER, using Lipofectamine 2000 (Invitrogen).

For reporter assays, NIH3T3 cells in 24-well plates (30000/well) were depleted of MRTF-A and MRTF-B by reverse transfection using Lipofectamine RNAiMax (Invitrogen) with the oligonucleotide 5'-UGGAGCUGGUGGAGAAGAA-3' (*Medjkane et al., 2009*). The day after, cells were transfected with 8 ng p3DA.luc SRF reporter, 20 ng Renilla reporter ptk-RL, and MRTF derivatives, or 2 ng C3

transferase. Reporter activity was measured by standard methods and luciferase activity expressed relative to renilla activity. Three technical replicates were performed per data point.

For gene expression analysis, RNA was prepared using GenElute (Sigma), reverse transcribed by superscript III (Invitrogen) and analysed by qPCR of intronic sequences using Express SYBRgreen (Invitrogen). Expression levels were expressed relative to GAPDH and expressed as mean ± SEM for three independent experiments. Primer sequences used were as follows: *Itgb1*, GAGTGGAAGCCC TGAAGACATT, TTGCCTTTTCCTTATGACTGACAA; *Myh9*, CATTTCCACATCGTGCTTCCTA, AGGG TTTTGGCACGTGTGA; *Vcl*, GATCCTGGTGTCTGTCGCTTCT, TGAGCAAAATGCCCCGAA; *Srf*, G TCAGGAATGGAGGATGGACAT, CCTTTCTCGGACTAGCACAGGTA; *Egr1,* GACCCAAACGTCCAG TCCTTTC, CAAGACCCTGGAGCTGTGTGAA; *Gapdh*, TCTTGTGCAGTGCCAGCCT, CCATA TGGCCAAATCCGTTCA.

## Antibodies, immunofluorescence and SDS-PAGE

Antibodies were Flag (F7425, Sigma-Aldrich), AlexaFluor 488 IgG (H+L) (Molecular probes, Eugene, OR), MRTF-A (C-19, Santa Cruz), panERK (BD Biosciences), Phospho-ERK1/2 (Cell Signalling), RhoA (Cell Signalling, 67B9), Crm1 and MLC2 (Santa Cruz) p-MLC2 (S19, Cell Signalling), rabbit, mouse and goat secondary IgG-HRP (DAKO). For MRTF-A phospho-specific antibodies, peptides were coupled to KLH and used to immunise rabbits. For immunoblots, sera were affinity-purified using the Sulfo-link kit (Pierce) or HiTrap-NHS columns (GE Healthcare), or used with unphosphorylated competing peptide. For assessment of changes in MRTF-A electrophoretic mobility by SDS-PAGE, 7% tris-acetate gels (Life Technologies) were run with tris-acetate running buffer and electrophoresis at 150V. Immunofluorescence was as described, with protein localisation scored as predominantly nuclear, pancellular or cytoplasmic (*Vartiainen et al., 2007*). For each sample, two replicates were scored, with at least 100 cells in each; error bars indicate SEM (n ≥ 3 independent experiments) or half-range (2 independent experiments). Localisation was scored as predominantly nuclear (N), pancellular (N/C), or predominantly cytoplasmic (C) as described. Statistical analysis used one- or two-way ANOVA, with Bonferroni multiple comparison test or post-hoc test, to evaluate proportions of (N+N/C) cells, except in cases where majority of base population was in this category, when proportions of (N) cells were compared. Data was processed using GraphPad Prism version 5.03.

## Mass spectrometry

NIH3T3 cells stably expressing Flag-MRTF-A were stimulated with serum for 30 min, immunoprecipitated with Flag antibody and fractionated by SDS-PAGE in triplicate. MRTF-A was excised, destained with 50% v/v acetonitrile and 50 mM ammonium bicarbonate, reduced with 10 mM DTT, and alkylated with 55 mM iodoacetamide before digestion with trypsin (Promega), AspN (Roche) or Chymotrypsin (Roche) overnight at 37°C. Analysis was with and without $TiO_2$ enrichment (Titansphere $TiO_2$ beads GLSciences) on the LTQ Orbitrap Discovery, operated in data-dependent mode to automatically switch between MS and MS/MS. Survey full scan MS spectra were acquired from m/z 350 to m/z 1500. The two (60-min method) or five (90-min method) most intense multiply charged precursor ions were fragmented in the linear ion trap using collisionally induced dissociation.

Raw mass spectrometric data were processed in MaxQuant (*Cox and Mann, 2008*) (version 1.3.0.5) for peptide and protein identification, and used for database search with the Andromeda search engine against the *Mus musculus* canonical sequence from UniProtKB release 2012_08. Fixed modifications were set as Carbamidomethyl (C) and variable modifications as oxidation (M) and phospho (STY). The estimated false discovery rate was set to 1% at the peptide, protein and site level. For all enzymatic digests a maximum of two missed cleavages was allowed. Other parameters were used as pre-set in the software. The summary table was generated from the MaxQuant output file PhosphoSTY Sites.txt, an FDR-controlled site-based table compiled by MaxQuant from the relevant information about the identified peptides.

## F- and G-actin quantitation

Cells were processed as described (*Vartiainen et al., 2007*) with modifications. G- and F-actin were detected using 0.3 µM DN'ase I-Alexa Fluor 488 conjugate and 5 U/ml Alexa Fluor 647-phalloidin (Molecular Probes). After three washes in PBS, cells were analysed using the Cellomics ArrayScan.

For each experiment, six technical replicates were performed. Statistical significance of alterations in F:G ratio was estimated by one-way ANOVA (Dunnett's multiple comparison test).

## Analysis of G-actin / MRTF complexes

Preparation of LatB-actin and fluorescence polarisation anisotropy assay were described (*Guettler et al., 2008*). Size exclusion chromatography was on Superdex 200 column equilibrated in 20 mM Tris (pH 8), 100 mM NaCl, 3 mM MgCl$_2$, 0.2 mM EGTA, 0.3 mM TCEP [(tris(2-carboxyethyl) phosphine)], 5% (v/v) glycerol, calibrated with globular proteins of known molecular weight.

## Acknowledgements

We thank Nicola O'Reilly (Crick Peptide Chemistry platform) for peptide synthesis; Koray Kirli and Dirk Görlich for running MRTF through their unpublished structure-based NES prediction programme; Michael Howell (Crick High Throughput Screening platform) and Graham Clark (Crick Equipment park) for technical support; Nick Totty for mass spectrometric analysis during the early stages of this project, and Bram Snijders for support; Shabaz Mohammed and Albert Heck (University of Utrecht) for access to the Orbitrap machine; and Axel Behrens, Thomas Güttler and Beric Henderson for plasmids; and members of the Signalling and Transcription group for helpful discussions and comments on the manuscript. This work was funded through core funding from CRUK to its London Research Institute until April 31st 2015, and thereafter by the Francis Crick Institute, which receives its core funding from Cancer Research UK (FC001-190), the UK Medical Research Council (FC001-190) and the Wellcome Trust (FC001-190); and by ERC Advanced Grant 268690 to RT. The authors have no conflicts of interest.

## Additional information

### Funding

| Funder | Grant reference number | Author |
| --- | --- | --- |
| Cancer Research UK | Crick Core Funding FC001-190 | Richard Panayiotou<br>Jessica Diring<br>Helen R Flynn<br>Richard Treisman |
| Medical Research Council | Crick Core Funding FC001-190 | Richard Panayiotou<br>Jessica Diring<br>Helen R Flynn<br>Richard Treisman |
| Wellcome Trust | Crick Core Funding FC001-190 | Richard Panayiotou<br>Jessica Diring<br>Helen R Flynn<br>Richard Treisman |
| European Research Council | Advanced Grant 268690 | Richard Panayiotou<br>Jessica Diring<br>Richard Treisman |
| Cancer Research UK | LRI Core Funding to 2015 | Richard Panayiotou<br>Francesc Miralles<br>Rafal Pawlowski<br>Jessica Diring<br>Helen R Flynn<br>Mark Skehel<br>Richard Treisman |

The funders had no role in study design, data collection and interpretation, or the decision to submit the work for publication.

### Author contributions

RPan, FM, RPaw, Experimental designs and interpretation, Designed and conducted molecular and cell biology experiments, Acquisition of data, Analysis and interpretation of data, Drafting or revising the article; JD, Experimental designs and interpretation, Designed and conducted molecular and cell

biology experiments, Acquisition of data, Analysis and interpretation of data; HRF, Experimental designs and interpretation, Acquisition of data, Analysis and interpretation of data, Drafting or revising the article; MS, Experimental designs and interpretation, Conducted mass spectrometric analysis, Acquisition of data, Analysis and interpretation of data; RT, Experimental designs and interpretation, Wrote the paper, Analysis and interpretation of data

### Author ORCIDs

Francesc Miralles, http://orcid.org/0000-0003-3069-2725
Helen R Flynn, http://orcid.org/0000-0001-7002-9130
Richard Treisman, http://orcid.org/0000-0002-9658-0067

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
