## [Decision Letter]

Thank you for submitting your article "Phosphorylation acts positively and negatively to regulate MRTF-A subcellular localisation and activity" for consideration by *eLife*. Your article has been favorably evaluated by Kevin Struhl as the Senior editor and three reviewers, one of whom is a member of our Board of Reviewing Editors.

The reviewers have discussed the reviews with one another and the Reviewing Editor has drafted this decision to help you prepare a revised submission

Summary:

This study of the MRTF-A transcription factor explores the role of phosphorylation in MRTF-A regulation. This is an important and interesting question. Previous studies have led to the identification of multiple sites of MRTF-A phosphorylation and there is no consensus in the field regarding the function of this phosphorylation.

The manuscript is presented in three sections. First, a mass spec analysis of epitope-tagged MRTF-A phosphorylation. The analysis presented complements existing knowledge (e.g. phosphosite). Moreover, the description of phosphospecific antibodies to some of these sites advances the field. The second and third sections of the manuscript focus on two sites of phosphorylation (Ser33 and Ser98) that regulate nucleocytoplasmic transport of MRTF-A. These sections of the manuscript provide insight into positive and negative regulation of MRTF-A nucleocytoplasmic transport. However, the use of MRTF-A fragments for the analysis – a well-justified reductionist approach – raises questions concerning the role of these phosphorylation sites in normal MRTF-A physiology.

Essential revisions:

1) The ERK-mediated phosphorylation of MRTF-A at Ser98 is interesting. This phosphorylation may reduce the stoichiometry of G-actin binding. Moreover, ERK may compete with G-actin for binding. It is proposed that disruption of G-actin binding promotes MRTF-A nuclear accumulation, based on studies of an MRTF-A fusion protein with PK. There are a number of open questions related to this section of the study. First, does ERK compete with G-actin binding? Second, what is the role of Ser98 phosphorylation in the normal physiology of MRTF-A? The S26A mutant is reported to undergo normal nucleocytoplasmic transport. It is therefore unclear what Ser98 phosphorylation might do. Studies of full-length MRTF-A are needed to establish the physiological significance of Ser98 phosphorylation.

2) The phosphorylation of Ser33 is interesting. Since it is not dependent on the ERK binding site identified for Ser98, this raises a question concerning the kinase that mediates phosphorylation at this site? Figure 2—figure supplement 2 should be expanded to include the pSer33 antibody to show the time course of pSer33 phosphorylation compared with the other sites. Figure 2—figure supplement 2 should be expanded to include the pSer33 antibody to show whether not phosphorylation at this site is sensitive to U0126. If the kinase is not ERK, what is it?

3) Ser33 phosphorylation is reported to regulate Crm1-mediated nuclear export. Data to support this conclusion is presented using a series of MRTF-A fragments. However, the role of Ser33 phosphorylation in normal MRTF-A physiology is unclear. Studies of full-length MRTF-A are needed to establish the physiological significance of Ser98 phosphorylation.

4) The text of the manuscript should be clarified to indicate whether the MRTF-G-actin interactions in question relate to those happening in the cytoplasm or to those taking place in the nucleus.

Essential revisions 1, 2 and 3 require the specific experiments that are stated. Essential revision 4 needs only text changes.

Essential revision #1 requires that they document the conclusion that ERK and G-actin compete for binding and that nucleocytoplasmic trafficking of full-length S98A MRTF-A (not just a fragment) is documented.

Essential revision #2 requires the requested additions to Figure 2—figure supplement 2.

Essential revision #3 requires that nucleocytoplasmic trafficking of full-length S33A MRTF-A (not just a fragment) is documented.

---

## [Author Response]

*Essential revisions:*

1) The ERK-mediated phosphorylation of MRTF-A at Ser98 is interesting. This phosphorylation may reduce the stoichiometry of G-actin binding. Moreover, ERK may compete with G-actin for binding. It is proposed that disruption of G-actin binding promotes MRTF-A nuclear accumulation, based on studies of an MRTF-A fusion protein with PK. There are a number of open questions related to this section of the study. First, does ERK compete with G-actin binding? Second, what is the role of Ser98 phosphorylation in the normal physiology of MRTF-A? The S26A mutant is reported to undergo normal nucleocytoplasmic transport. It is therefore unclear what Ser98 phosphorylation might do. Studies of full-length MRTF-A are needed to establish the physiological significance of Ser98 phosphorylation.

ERK-MRTF interactions: We were unable to document competition between G-actin and ERK in pulldown experiments. However, in a more functionally relevant approach we were able to show that G-actin inhibits phosphorylation of S98 by recombinant ERK in vitro. Thus G-actin binding and ERK phosphorylation are mutually antagonistic.

Experiment shown in new Figure 4, text in the first paragraph of the subsection “ERK-mediated S98 phosphorylation and G-actin binding are mutually inhibitory”

Full length proteins: for discussion, see point 3 below.

2) The phosphorylation of Ser33 is interesting. Since it is not dependent on the ERK binding site identified for Ser98, this raises a question concerning the kinase that mediates phosphorylation at this site? Figure 2—figure supplement 2 should be expanded to include the pSer33 antibody to show the time course of pSer33 phosphorylation compared with the other sites. Figure 2—figure supplement 2 should be expanded to include the pSer33 antibody to show whether not phosphorylation at this site is sensitive to U0126. If the kinase is not ERK, what is it?

Comment: Absence of docking-site dependence does not necessarily indicate ERK independence, since S33 phosphorylation may involve a distinct docking site. For example, in Elk-1, S383 phosphorylation is critically dependent on the FQFP motif but not the D-box motif (Fantz et al. JBC 276 25926 (2001).

Response: S33 phosphorylation is increased only slightly by serum or TPA stimulation, but in a U0126-sensitive fashion, indicating the involvement of ERK (new Figure 4; Figure 2—figure supplement 2). Consistent with this, S33 phosphorylation can also be elevated by expression of constitutively active MEK (Figure 4—figure supplement 1). Thus ERK must make at least some contribution to S33 phosphorylation, although other kinases may be involved, as there is a high basal level of S33 phosphorylation in resting cells. Recent studies with the EKAREV ERK biosensor suggest that S/P phosphorylation may CDK-dependent (Aoki et al., Mol Cell 52, p529 (2013). Note added in the second paragraph of the subsection “ERK-mediated S98 phosphorylation and G-actin binding are mutually inhibitory”.

New Figure 2—figure supplement 2 include S33 phosphorylation in response to FCS, CD and TPA, all ± U0126. See also revised text in the aforementioned paragraph.

3) Ser33 phosphorylation is reported to regulate Crm1-mediated nuclear export. Data to support this conclusion is presented using a series of MRTF-A fragments. However, the role of Ser33 phosphorylation in normal MRTF-A physiology is unclear. Studies of full-length MRTF-A are needed to establish the physiological significance of Ser98 phosphorylation.

Effects of S98 and S33 mutations on the full length MRTF-A are now added in a new Figure 5, which suggests that the sites contribute to regulation in a similar way to their behaviour in the fragment assay. New text was also added to the last paragraph of the subsection “MRTF-A N-terminal phosphorylations affect nucleocytoplasmic shuttling”. New Figure 7, extend this analysis by demonstrating that S98 and S33 cooperate with each other and with the NES elements to set the baseline level of cytoplasmic MRTF in resting cells. New text was added to the last paragraph of the subsection “The N-terminal NES cooperates with multiple MRTF NES elements”.

New Figure 5 shows that:

a) S98A blunts nuclear accumulation in response to TPA, consistent with its behaviour of the MRTF fragment assay; surprisingly S98D also slightly inhibits accumulation, but the small effect hard to interpret in the full-length protein;

b) S33A raises proportion of resting cells with nuclear MRTF-A and doesn't really affect TPA induction; while S33D blunts induction. Both these results are consistent with the results observed with the fragments.

We return to the roles of S33 and S98 phosphorylation when we analyse the multiple NES elements in Figure 7. New Figure 7 shows that S33A cooperates with NES2 and NES4 to maintain full-length MRTF-A in the cytoplasm in resting cells; it works almost as well as inactivation of NES1, while S33D has little effect. Similarly, 98D cooperates with NES2, NES4 in this assay, and also with S33A, similar to the fragment assay.

*4) The text of the manuscript should be clarified to indicate whether the MRTF-G-actin interactions in question relate to those happening in the cytoplasm or to those taking place in the nucleus.*

Our previous FRET-FLIM studies (Vartiainen et al. Science 316, p1749 (2007)) showed that actin / MRTF-A interaction can be detected both in the cytoplasm and the nucleus. Importantly, even when MRTF-A is confined to the nucleus by inactivation of Crm1, regulated actin-MRTF interaction can be detected. Thus, actin- MRTF-A interaction occurs in both nucleus and cytoplasm. Subsequent studies have shown that specific disruption of nuclear actin-MRTF-A interaction can be sufficient to activate MRTF-A (Baarlink et al., Science 340, p864 (2013). Perhaps not surprisingly, in stimulated cells most S98-phosphorylated MRTF-A is nuclear, so it is most likely that its primary point of action is in the nucleus.

Text reworded in the Introduction, second paragraph; Results, subsection “G-actin binding negatively regulates MRTF phosphorylation”, last paragraph; new Figure 4—figure supplement 1; Discussion, subsection “G-actin binding controls MRTF-A phosphorylation”, last paragraph; and end of subsection “G-actin binding controls MRTF-A phosphorylation”.